# MMedAgent-RL: Optimizing Multi-Agent Collaboration for Multimodal Medical Reasoning

**Peng Xia[1,2†,\*] Jinglu Wang[2†], Yibo Peng[2,3†\*], Kaide Zeng[1], Zihan Dong[4], Xian Wu,**
**Xiangru Tang[5], Hongtu Zhu[1], Yun Li[1], Linjun Zhang[4], Shujie Liu[2], Yan Lu[2‡], Huaxiu Yao[1‡]**
[1]UNC-Chapel Hill, [2]Microsoft Research, [3]CMU, [4]Rutgers University, [5]Yale University
`{pxia,huaxiu}@cs.unc.edu, {jinglu.wang,yanlu}@microsoft.com`

## Abstract

Medical Large Vision-Language Models (Med-LVLMs) have shown strong potential in multimodal diagnostic tasks. However, existing single-agent models struggle to generalize across diverse medical specialties, limiting their performance. Recent efforts introduce multi-agent collaboration frameworks inspired by clinical workflows, where general practitioners (GPs) and specialists interact in a fixed sequence. Despite improvements, these static pipelines lack flexibility and adaptability in reasoning. To address this, we propose MMedAgent-RL, a reinforcement learning (RL)-based multi-agent framework that enables dynamic, optimized collaboration among medical agents. Specifically, we train two GP agents based on Qwen2.5-VL via RL: the triage doctor learns to assign patients to appropriate specialties, while the attending physician integrates the judgments from multi-specialists and its own knowledge to make final decisions. To address the inconsistency in specialist outputs, we introduce a curriculum learning (CL)-guided RL strategy with dynamic entropy regulation, progressively teaching the attending physician to balance between imitating specialists and correcting their mistakes. Experiments on five medical VQA benchmarks demonstrate that MMedAgent-RL outperforms both open-source and proprietary Med-LVLMs. Notably, it achieves an average performance gain of 23.6% over strong baselines.

## 1 Introduction

Large Vision-Language Models (LVLMs) are becoming increasingly proficient in visual understanding and reasoning (Liu et al., 2024a;b; Zhu et al., 2023; Bai et al., 2023; Chen et al., 2024c). This advancement is also making a significant impact in the biomedical domain, where Medical Large Vision-Language Models (Med-LVLMs) have demonstrated great potential in enabling intelligent diagnostic applications (Li et al., 2023; Moor et al., 2023; Nath et al., 2024). However, as shown in Figure 1 (a) *left*, although a single Med-LVLM can be trained with a large amount of data and show promise results to some extent, it is challenging for a single model to handle diagnostic expertise from different subfields (e.g., radiology, pathology, etc.).

Therefore, some recent works propose using multi-agent collaboration to solve medical tasks (Li et al., 2024c; Kim et al., 2024; Tang et al., 2024), where different models act as specialists or general practitioners, collaborating and discussing to arrive at a final answer, improving overall performance compared to a single agent. These works follow the steps of simulating a hospital visit process and adopt a General Practitioner (GP) $\rightarrow$ Specialist $\rightarrow$ GP workflow. First, the general practitioner (i.e., the *triage doctor*) classifies the patient based on the consultation questions and images and selects the appropriate department from several predefined specialties. Then, *specialist doctors* from the relevant departments provide their diagnoses. Finally, the general practitioner (i.e., *attending physician*) makes the final decision based on the images, consultation questions, and the diagnostic results from multiple specialists. However, as illustrated in Figure 1 (a) *middle*, such workflows are inherently *static*. Such interaction pattern between agents is fixed and predetermined, which limits the system's capacity for flexible, optimized reasoning across multiple modalities.

---

*Work done at Microsoft Research. †Equal Contribution. ‡Corresponding Authors.

To address this challenge, motivated by the success of Reinforcement Learning (RL)-driven reasoning (Jaech et al., 2024; Guo et al., 2025; Team et al., 2025), as shown in Figure 1 (a) *right*, we perform a stage-wise training on two GPs based on Qwen2.5-VL (Bai et al., 2025) via reinforcement learning, namely the triage doctor and the attending physician.

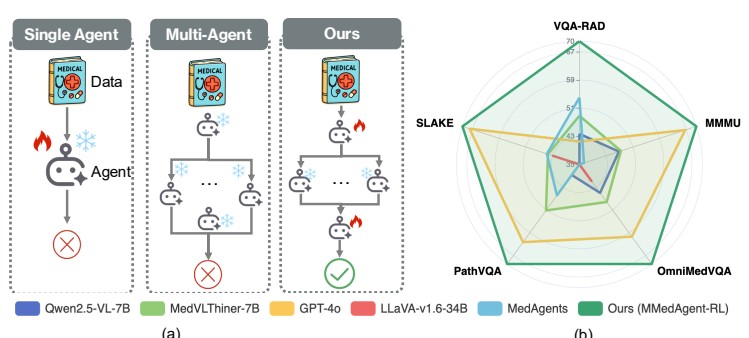

Specifically, first, for the first GP, i.e., triage doctor, we utilize the image modality information provided by the dataset itself to reinforce the triage doctor, such as pathology slides → *Pathologist*, ensuring that the triage doctor can accurately assign patients to the appropriate department. Then, we use powerful proprietary models like GPT (OpenAI, 2025) to play the role of the specialist doctors and generate initial judgments. Finally, the second GP, i.e., attending physician, integrates domain knowledge from multiple specialists and their own judgment to make the final decision. Here, during the process of the general practitioner integrating specialist doctor information, while specialist doctors provide valuable domain knowledge, their judgments are not always perfectly accurate. These inconsistencies in specialist performance can introduce noise into model training, preventing the model from simply memorizing or blindly replicating their outputs. Instead, the model must learn to generalize beyond potentially flawed expert judgments.

Figure 1: Comparison of Med-Agent paradigms: single-agent → static workflows → dynamic collaboration. (a) Motivation: Single-agent models struggle with domain specialization, and prior multi-agent systems rely on fixed workflows, limiting adaptability. We propose a trainable reasoning-enhanced multi-agent system via RL. (b) Performance: Our method is highly competitive across multiple benchmarks.

To address this, inspired by Curriculum Learning (CL) (Bengio et al., 2009; Pentina et al., 2015; Deng et al., 2025), which enables models to be trained progressively on increasingly difficult tasks, we implement an entropy-aware reinforcement learning approach based on CL, aiming to help the model gradually learn to leverage the knowledge of specialist doctors. The core principle is that the attending physician agent faces a dynamic *exploration-exploitation dilemma* when integrating specialist opinions: it must learn when to trust and adopt a consensus and when to challenge it and search for novel diagnostic paths (Cui et al., 2025; Wang et al., 2025b). Specifically, we use the accuracy of specialist results as a flag to classify the training data by difficulty: specialist results that are *completely correct are labeled as easy*, *partially correct as medium*, and *completely incorrect as hard*. In this way, we design a three-stage curriculum reinforcement learning process for optimizing the attending physician to handle diverse specialist results, where the model starts with low-entropy policies to exploit reliable "easy" cases and gradually increases its policy entropy to explore and correct flawed judgments in "hard" cases.

The primary contribution of this paper is MMedAgent-RL, an RL-driven framework optimized for multi-agent collaboration in improving medical reasoning. Empirical results on five medical multimodal datasets, shows that the model performs exceptionally well not only on in-domain datasets but also on out-of-domain datasets, outperforming a series of both open-source and proprietary LVLMs, exceeding SFT method by 23.6%.

## 2 PRELIMINARIES

In this section, we will provide a brief overview of LVLMs, multi-agent collaboration and GRPO.

**Large Vision Language Models**. LVLMs enhance LLMs by integrating visual input $x_v$ with textual input $x_t$, forming a joint input $x = (x_v, x_t)$. They autoregressively predict the next token's distribution to generate a textual response $y$.

**Multi-Agent Collaboration**. To support complex workflows, multi-agent frameworks coordinate specialized agents. Our setting simulates a hospital visit: GP → Specialist(s) → GP. Each agent

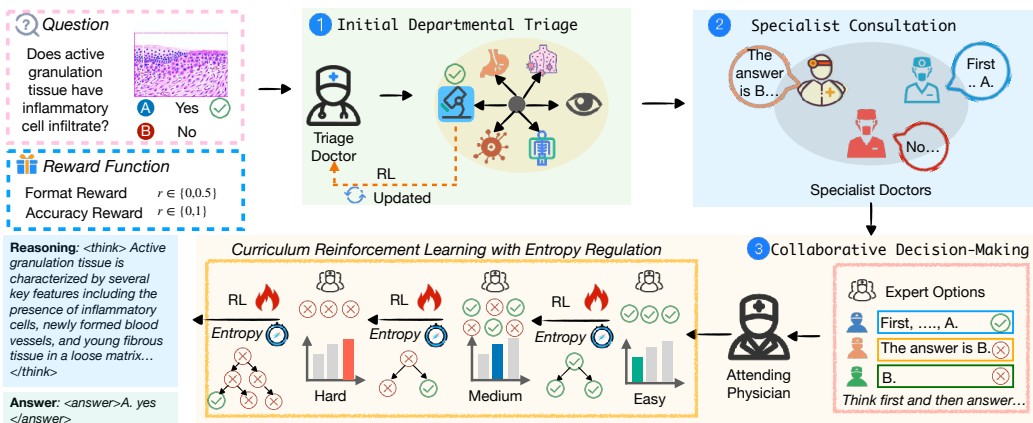

Figure 2: Overview of MMedAgent-RL, a RL-driven multi-agent framework designed for multimodal medical reasoning. It simulates the clinical loop of General Practitioner (GP) → Specialists → GP. First, MMedAgent-RL optimizes the triage doctor (the first GP) to improve triage accuracy. Then, proprietary LVLMs are used as the specialist doctors for the assigned department. Finally, curriculum learning and RL are combined to progressively train the attending physician (the second GP), who integrates the specialist knowledge and makes robust decisions.

$a_i \in \mathcal{A}$ follows policy $\pi_{\theta_i}(y \mid x)$, with multimodal input $x = (x_v, x_t)$, where $x_v$ is an image, $x_t$ is a text instruction, and $y$ is the output. GP agent: $a_{\text{GP}}$; specialists: $\{a_{\text{SP}}^{(1)}, \ldots, a_{\text{SP}}^{(K)}\}$. The workflow proceeds as follows: 1) Triage: $a_{\text{GP}}^{\text{triage}}$ selects department via $d = \arg\max_k \pi_{\theta_{\text{GP}}^{\text{triage}}}(k \mid x)$. 2) Specialist: $a_{\text{SP}}^{(d)}$ produces response $y_d \sim \pi_{\theta_{\text{SP}}^{(d)}}(y \mid x)$. 3) Aggregation: $a_{\text{GP}}^{\text{attend}}$ outputs $y_{\text{final}} \sim \pi_{\theta_{\text{GP}}^{\text{attend}}}(y \mid x, y_d)$.

**Group Relative Policy Optimization (GRPO).** Group Relative Policy Optimization (GRPO) (Guo et al., 2025) is a reinforcement learning method that avoids training a critic by using intra-group relative rewards to optimize the policy. For each query $x$, the model samples $G$ responses $\{y^{(1)}, \ldots, y^{(G)}\}$, which are scored to get rewards $\{R_1, \ldots, R_G\}$. GRPO computes normalized advantages and updates the policy with a PPO-style clipped objective (Schulman et al., 2017):

$$\mathcal{J}_{\text{GRPO}}(\theta) = \mathbb{E}_{x, \{y_i\}} \left[ \frac{1}{G} \sum_{i=1}^{G} \left( \min\left(r_i A_i, \text{clip}(r_i, 1-\epsilon, 1+\epsilon) A_i\right) - \beta \, \mathbb{D}_{\text{KL}}(\pi_\theta \| \pi_{\text{ref}}) \right) \right], r_i = \frac{\pi_\theta(y_i \mid x)}{\pi_{\text{old}}(y_i \mid x)}, \tag{2.1}$$

where $A_i = \frac{R_i - \text{mean}(\{R_j\}_{j=0}^G)}{\text{std}(\{R_j\}_{j=0}^G)}$, $\epsilon$, $\beta$ are hyperparams, and $\pi_{\text{old}}$ is a old policy model. GRPO enables scalable policy learning using only relative rewards, without a critic.

# 3 METHODOLOGY

In this section, as illustrated in Figure 2, we will present MMedAgent-RL, a RL-driven multi-agent framework for multimodal medical reasoning by emulating a structured clinical workflow. Our approach begins with the first General Practitioner (GP) leveraging the input to select the appropriate medical department for further consultation. To optimize the initial triage decision, we employ GRPO to refine the triage doctor's capabilities. Then the case is referred to a panel of specialist doctors, each served by a proprietary LVLM specialized in the identified department, analyze data and provide expert opinions. Finally, the process culminates with the second GP, acting as the attending physician, integrates all specialist opinions to form a final diagnosis. We will delve into each stage as follows:

## 3.1 INITIAL DEPARTMENTAL TRIAGE

In real-world medical treatment, particularly for complex diagnoses, the workflow fundamentally relies on a Multi-Disciplinary Team (MDT) process (Abo-Hamad & Arisha, 2013): moving from initial assessment and triage to expert consultation, and finally, synthesis by an attending physician. This collaborative paradigm has been validated in recent medical AI literature as an effective abstraction for simulating clinical decision-making. Previous works like Agent Hospital (Li et al., 2024c) and

MDAgents (Kim et al., 2024) have adopted similar role-based structures, employing LLMs for triage or dynamic collaboration. However, these methods typically rely on rigid or predefined assignment strategies and lack the ability to update the model based on new data.

To address this challenge, the first step is to optimize the general practitioner $a_{\mathrm{GP}}^{\mathrm{triage}}$ who acts as the triage doctor (i.e., policy $\pi_{\theta_{\mathrm{GP}}^{\mathrm{triage}}}$). Here, we use the image modality information provided by the dataset itself as ground truth labels $y^*$ to train the triage model. For example, pathology slides $\rightarrow$ pathologist (e.g., PathVQA contains pathology slide images and is thus assigned to pathologists), ensuring that the triage model can accurately assign patients to the appropriate medical specialty.

Specifically, when prompting the triage doctor, we provide $k$ candidate specialties. In our setup, $k$ is set to 7, including *Pathologist, Radiologist, Surgeon, Oncologist, Endocrinologist, Ophthalmologist, and Dermatologist*, which broadly cover the main departments involved in the data. Our aim is not only to improve triage accuracy, but also to strengthen the model's reasoning process, helping explain why a particular triage recommendation is made. Thus, we use GRPO as the base RL algorithm. At this stage, the reward function adopts a rule-based format with rewards $R_{\mathrm{format}} \in \{0, 0.5\}$ and accuracy rewards $R_{\mathrm{accuracy}} \in \{0, 1\}$. A reward is given when the output format meets the required criteria and the chosen specialty is correct. This stage can optimize the triage doctor's performance, improving their ability to select the appropriate specialty $d = \arg\max_k \pi_{\theta_{\mathrm{GP}}^{\mathrm{triage}}}(k \mid x)$, and lays a foundation for subsequently acting as the corresponding specialist.

## 3.2 ROLE-PLAYING SPECIALISTS OFFER VALUABLE INSIGHTS

After obtaining the department from the triage doctor, following previous work using LLMs or LVLMs for medical discussions (Li et al., 2024c; Tang et al., 2024; Li et al., 2024a), we utilize several powerful models as specialist doctors $a_{\mathrm{SP}}^{(d)}$ to provide relatively accurate preliminary judgments. This facilitates subsequent reference by the attending physician. In our setup, we use responses from $e$ specialists as references for each sample. We only require the specialist doctors to independently provide expert opinions $y_d \sim \pi_{\theta_{\mathrm{SP}}^{(d)}}(y \mid x)$ within their specialty, without engaging in complex interactions. This ensures system efficiency and avoids majority voting that could overshadow minority opinions, leaving the final decision to the attending physician.

## 3.3 EVOLVING DECISIONS BY ATTENDING PHYSICIAN VIA ONGOING COLLABORATION

After getting the responses from the specialists, we then integrate their knowledge into the final general practitioner designed to support the final diagnostic decision. The final decision-making agent, namely the attending physician, plays the most crucial role throughout the diagnostic process, as they must synthesize diverse expert opinions and draw upon their own clinical expertise to arrive at a final judgment. This poses significant challenges for the attending physician, as the specialists' conclusions are not always fully reliable. Over-reliance on specialist input can lead to suboptimal outcomes. Secondly, different specialists may offer conflicting interpretations of the same case, creating misalignment issues. If the model is unable to reconcile its internal reasoning with external expert input, it risks compounding errors. For instance, while majority voting may help mitigate the influence of less competent specialists, it can also suppress minority views, including potentially the only correct one. Such multi-agent collaboration can yield adverse effects when the model is not properly aligned with the nature and limitations of expert knowledge.

To address these challenges, we draw inspiration from curriculum learning (Bengio et al., 2009), which emphasizes the importance of organizing learning experiences in a meaningful progression, i.e., from easier to harder tasks. Motivated by this principle, see in Alg. 1, we propose the Curriculum-based Entropy-Aware Reinforcement Learning for Multi-Agent Collaboration (**C-MARL**). The core principle of this framework is that the attending physician agent faces a dynamic *exploration-exploitation dilemma* when integrating specialist opinions: it must learn when to trust and adopt a consensus (exploit) and when to challenge it and search for novel diagnostic paths (explore). We posit that the policy entropy is the central mechanism for navigating this trade-off (Cui et al., 2025). Therefore, C-MARL employs a purpose-designed curriculum to actively shape the policy entropy of the attending physician, enabling it to adapt to specialist information of varying reliability.

---

**Algorithm 1:** Curriculum-Based Multi-Agent Reinforcement Learning (C-MARL)

---

**Input:** Dataset $\mathcal{D} = \{x_v^{(i)}, x_t^{(i)}, y^{*(i)}\}_{i=1}^N$, policy model $\pi_\theta$, old policy $\pi_{\text{old}}$, group size $G$, specialist responses $y_d^{(i)}$.

**Output:** $\pi_\theta$.

1 Initialize $\mathcal{D}_{\text{easy}}, \mathcal{D}_{\text{medium}}, \mathcal{D}_{\text{hard}}$ as empty sets

2 **foreach** $(x_v, x_t, y^*) \in \mathcal{D}$ **do**

3     $\triangleright$ *Use Specialists' Accuracy to Categorize the Dataset by Task Difficulty*

4     Calculate the accuracy of the specialist doctor $s \leftarrow \text{Acc}(y_d, y^*)$

5     **if** $s = 1$ **then**

6         Put $\{(x_v, x_t), y^*\}$ into $\mathcal{D}_{\text{easy}}$

7     **if** $0 < s < 1$ **then**

8         Put $\{(x_v, x_t), y^*\}$ into $\mathcal{D}_{\text{medium}}$

9     **if** $s = 0$ **then**

10         Put $\{(x_v, x_t), y^*\}$ into $\mathcal{D}_{\text{hard}}$

11 **foreach** $(x_v, x_t, y^*) \in \{\mathcal{D}_{easy}, \mathcal{D}_{medium}, \mathcal{D}_{hard}\}$ *in batch* **do**

12     $\triangleright$ *Utilize the RL Algorithm for Optimization at Each Stage*

13     Sample $G$ rollouts $\{y_{\text{final}}^{(1)}, y_{\text{final}}^{(2)}, \cdots, y_{\text{final}}^{(G)}\}$ from $\pi_{\text{old}}$, where $y_{\text{final}}^{(g)} \leftarrow \pi_\theta(y \mid (x_v, x_t), y_d)$

14     **foreach** *rollout $y_{final}$* **do**

15         Calculate the outcome reward $R(y_{\text{final}}) = R_{\text{format}}(y_{\text{final}}) + R_{\text{accuracy}}(y_{\text{final}})$

16     Compute the groupwise advantage $A_i \leftarrow \frac{R_i - \text{mean}(\{R_j\}_{j=0}^G)}{\text{std}(\{R_j\}_{j=0}^G)}$

17     Calculate the entropy regularization term $H_t \leftarrow -\sum_{j=1}^V p_{t,j} \log p_{t,j}$

18     Compute the loss in Equation 3.1 and update $\pi_\theta$

---

**Curriculum Design based on Specialist Reliability**. Unlike previous curriculum learning approaches that define difficulty based on problem formulation or data domains, we categorize tasks based on the accuracy of specialists' diagnoses $y_d \sim \pi_{\theta_{\text{SP}}^{(d)}}(y_d \mid x)$, denoted by $s = \text{Acc}(y_d, y^*)$. The dataset is divided into three levels: *fully correct specialist results ($s = 1$) are labeled as easy*, *partially correct results ($0 < s < 1$) are labeled as medium*, and *completely incorrect results ($s = 0$) are labeled as hard*. The datasets corresponding to the three levels are denoted as $\mathcal{D}_{\text{easy}}, \mathcal{D}_{\text{medium}}, \mathcal{D}_{\text{hard}}$, respectively, and $\mathcal{D} = \mathcal{D}_{\text{easy}} \cup \mathcal{D}_{\text{medium}} \cup \mathcal{D}_{\text{hard}}$. Based on these data of three categories, we design a three-stage curriculum reinforcement learning process to optimize the attending physician's ability to handle different types of specialist knowledge, such as when to accurately leverage specialist knowledge and when to rely on their own understanding to solve problems.

**Reinforcement Learning with Dynamic Entropy Regulation**. We utilize GRPO as our base RL algorithm. For each query $x$, the attending physician $\pi_{\theta_{\text{GP}}^{\text{attend}}}$ generates a group of $G$ responses $\{y_{\text{final}}^{(i)}\}_{i=1}^G$. The reward $R_i$ for each response is determined by a format reward $R_{\text{format}} \in \{0, 0.5\}$ and an accuracy reward $R_{\text{accuracy}} \in \{0, 1\}$, and GRPO calculates the relative advantage $A_i = \frac{R_i - \text{mean}(\{R_j\})}{\text{std}(\{R_j\})}$ for the policy update. To achieve dynamic entropy control, we introduce an entropy regularization term into the standard GRPO objective function:

$$\mathcal{J}_{\text{C-MARL}}(\theta) = \mathbb{E}\left[\mathcal{J}_{\text{GRPO}}(\theta) + \gamma_s \cdot H_t(\pi_{\theta_{\text{GP}}^{\text{attend}}})\right], \quad H_t = -\sum_{j=1}^V p_{t,j} \log p_{t,j}, \quad (3.1)$$

$$\text{where} \quad \mathbf{p}_t = \pi_\theta(\cdot \mid \mathcal{R}_{<t}, x; T) = \text{Softmax}\left(\frac{\mathbf{z}_t}{\tau}\right).$$

Here, $\mathcal{J}_{\text{GRPO}}$ is the PPO-clip loss term from GRPO in Eq. 2.1, $V$ is the vocabulary size, $\mathbf{z}_t \in \mathbb{R}^V$ are the pre-softmax logits, and $\tau$ is the decoding temperature. Critically, the entropy bonus coefficient $\gamma_s$ is not a fixed hyperparameter but is dynamically set based on the curriculum level $s$ of the current sample. For $s = 1$, we set $\gamma_{\text{easy}} \approx 0$, as we do not need to explicitly reward exploration when the agent should be learning to exploit reliable specialist knowledge. For $0 < s < 1$, we use a moderate positive bonus $\gamma_{\text{medium}} > 0$ to encourage policy diversity and prevent the agent from becoming overconfident in the face of conflicting information. For $s = 0$, we apply a strong positive bonus $\gamma_{\text{hard}} \gg \gamma_{\text{medium}}$ to aggressively incentivize exploration, compelling the model to break from the misleading specialist consensus.

## 4 THEORETICAL ANALYSIS

In this section, we provide theoretical insights explaining why curriculum learning works better than the usual SGD when we optimize the GRPO objective function 3.1. Specifically, we begin by analyzing the effectiveness of the curriculum learning strategy for policy learning. Our goal is to determine the number of samples and iterations required to achieve a specified error tolerance.

The curriculum learning procedure can be simplified as follows. We consider $J$ batches of samples arranged in increasing order of difficulty. In our setting (Section 3.2), the curriculum is composed of $J = 3$ difficulty stages: easy, medium, and hard. Starting from $j = 1$, we sequentially train on each batch using Stochastic Gradient Descent (SGD) to obtain the policies $\pi_{\widehat{\theta}_{j,K_j}}(y_i \mid x), j = 1, \ldots, J$.

We aim to track the policy's trajectory and convergence throughout this process. To model the difficulty of the data from each batch, we make the following assumption and suppose the target parameter $\theta^* = \theta_J^*$:

**Assumption 4.1.** *From the easiest to the hardest dataset, for $j$-th dataset, we assume*

$$\mathcal{D}_j = \left\{ (x_i, y_i)_{i=1}^{n_j} \mid x_i \sim p(x), \ y_i \sim \pi_{\theta_j^*}(y \mid x) \right\}$$

**Theorem 4.1** (Informal). *Suppose Assumption 4.1 holds, together with some regularity conditions (Assumptions B.2–B.6 in Appendix B). Take $\epsilon_1 < L_1 \delta^2 / 16$. For curriculum learning, in order to make the error $\|\widehat{\theta}_{\text{cl}} - \theta^\star\|_2^2 < \frac{4\epsilon_1}{L_1}$, set $n_j$ satisfies $\mathcal{R}_{n_j}(\Pi) \leq \epsilon_1/4$, where $\mathcal{R}_{n_j}(\Pi)$ is the Rademacher complexity of policy class $\Pi$ and uses the learning rate $\eta \leq \mu/L_2^2$, and set the total iteration as*

$$K = \Omega\left( \frac{1}{\mu\eta} \sum_{j=0}^{J-1} \log \frac{L_2^2 \|\theta_j^\star - \theta_{j+1}^\star\|_2^2}{\mu \, \epsilon_1} \right).$$

*For regular SGD, in order to make the error $\|\widehat{\theta}_{\text{rg}} - \theta_{\text{rg}}^\star\|_2^2 < \frac{4\epsilon_1}{L_1}$, set $n_{\text{rg}}$ to satisfy $\mathcal{R}_{n_{\text{rg}}}(\Pi) \leq \epsilon_1/4$ and uses the same $\eta$, and take*

$$K_{\text{rg}} = \Omega\left( \frac{1}{\mu\eta} \log \frac{L_2^2 \|\theta_0^\star - \theta_{\text{rg}}^\star\|_2^2}{\mu \, \epsilon_1} \right).$$

*With probability at least $1 - \exp\left( -\frac{n_{\text{rg}} \epsilon_1^2}{2B_1^2} \right) - \sum_{j=1}^{J} \exp\left( -\frac{n_j \epsilon_1^2}{2B_1^2} \right) - (J+1)\epsilon_1$, we have*

$$\|\widehat{\theta}_{\text{cl}} - \theta^\star\|_2^2 < \|\widehat{\theta}_{\text{rg}} - \theta^\star\|_2.$$

*Here, $\theta_0^\star$ is the starting point; $\widehat{\theta}_{\text{cl}}$ and $\widehat{\theta}_{\text{rg}}$ are the final outputs of curriculum learning and standard SGD, respectively; $n_j$ is the sample size of batch $j$; $n_{\text{rg}}$ is the total sample size for standard SGD; $\delta$ measures the distance between the true minimizer $\theta^*$ and the population minimizer of standard SGD $\theta_{\text{rg}}^\star$; $B_1$, $L_1$, $L_2$, $\mu$, $\Omega(\cdot)$ and $\mathcal{R}_n(\cdot)$ are defined in Appendix B.*

*Remark* 4.1. This theorem provides a formal justification for the convergence of curriculum learning. The total training time, $K$, depends on the sum of logarithmic distances between the optimal policies of consecutive stages: $\sum_{j=0}^{J-1} \log \|\theta_j^\star - \theta_{j+1}^\star\|_2^2$. An effective curriculum ensures these intermediate distances are small, allowing the solution from each stage to serve as a strong **warm start** for the next. CL thereby decomposes a single, challenging optimization problem, leaping directly from an initial $\theta_0^\star$ to the final $\theta_J^\star$, into a sequence of more tractable sub-problems, creating an efficient optimization path. In contrast, we establish a lower bound for standard SGD which shows that, under these conditions, it fails to converge to the optimal policy. Details are provided in Appendix B.

## 5 EXPERIMENTS

In this section, we evaluate the performance of MMedAgent-RL, aiming to answer the following questions: (1) Can MMedAgent-RL effectively improve model performance compared to other LVLMs and the Qwen2.5-VL-based baselines? (2) How does MMedAgent-RL perform on out-of-distribution datasets? (3) Does each proposed component contribute to performance gains? (4) What is the impact of choosing different models as specialist doctors? (5) Does MMedAgent-RL truly enhance the model's capabilities across various specialist configurations?

Table 1: The results of the medical VQA benchmark. Here, MMMU denotes MMMU (Health & Medicine track). The best results and second best results are highlighted in red and blue, respectively. Majority voting is used for the test-time scaling (TTS).

| Model | In-Domain Datasets | | | | Out-of-Distribution Datasets | | |
|---|---|---|---|---|---|---|---|
| | VQA-RAD | SLAKE | PathVQA | Avg. | OmniMedVQA | MMMU-Med | Avg. |
| GPT-4o | 61.0 | 75.5 | 69.4 | 68.6 | 68.5 | 69.7 | 69.1 |
| Med-Flamingo | 45.4 | 43.5 | 54.7 | 47.9 | 30.7 | 28.3 | 29.5 |
| RadFM | 50.6 | 34.6 | 38.7 | 41.3 | 28.2 | 27.0 | 27.6 |
| LLaVA-Med-7B | 51.4 | 48.6 | 56.8 | 52.3 | 44.1 | 36.9 | 40.5 |
| Qwen-VL-Chat | 47.0 | 56.0 | 55.1 | 52.7 | 48.3 | 32.7 | 40.5 |
| Yi-VL-34B | 53.0 | 58.9 | 47.3 | 53.1 | 51.5 | 41.5 | 46.5 |
| LLaVA-v1.6-7B | 52.6 | 57.9 | 47.9 | 52.8 | 49.0 | 33.1 | 41.1 |
| LLaVA-v1.6-13B | 55.8 | 58.9 | 51.9 | 55.5 | 48.0 | 39.3 | 43.7 |
| LLaVA-v1.6-34B | 58.6 | 67.3 | 59.1 | 61.6 | 58.7 | 48.8 | 53.8 |
| LLaVA-v1.5-LLaMA3-8B | 54.2 | 59.4 | 54.1 | 55.9 | 44.6 | 38.2 | 41.4 |
| HuatuoGPT-Vision-7B | 63.0 | 77.2 | 58.7 | 66.3 | 74.6 | 51.0 | 62.8 |
| Qwen2.5-VL-3B | 61.0 | 62.7 | 57.6 | 60.4 | 60.1 | 54.5 | 57.3 |
| Qwen2.5-VL-7B | 61.8 | 64.7 | 60.5 | 62.3 | 60.8 | 56.6 | 58.7 |
| MedVLThinker-7B | 63.7 | 67.8 | 65.2 | 65.6 | 62.4 | 57.0 | 59.7 |
| **Multi-Agent Collaboration** | | | | | | | |
| MedAgents | 65.6 | 67.9 | 63.2 | 65.6 | 55.8 | 49.7 | 52.6 |
| MDAgents | 66.8 | 68.2 | 65.4 | 66.8 | 58.2 | 52.3 | 55.1 |
| AFlow | 67.3 | 68.9 | 66.4 | 67.5 | 59.6 | 53.6 | 56.6 |
| **MMedAgent-RL** (7B) | 71.5 +10% | 76.2 +12% | 72.3 +12% | 73.3 +11% | 73.3 +13% | 71.9 +15% | 72.6 +14% |
| *w/ Test-Time Scaling* | 73.9 +12% | 80.1 +15% | 74.3 +14% | 76.1 +14% | 79.6 +19% | 73.5 +17% | 76.6 +18% |

## 5.1 EXPERIMENTAL SETUP

**Implementation Details.** We use Qwen2.5-VL (Bai et al., 2025) as the base model. We design the prompt template shown in Table 4, clearly specifying the required output structure, which includes using <think> and <answer> tags to separately contain the reasoning process and the final answer. The rollout batch size and training batch size are both set to 128, with 8 rollouts generated for each sample. The sampling temperature is set to 1.0 to encourage response diversity, and optimization is done with a learning rate of $1 \times 10^{-6}$. The KL divergence coefficients are set to $1 \times 10^{-3}$, $4 \times 10^{-3}$, and $1 \times 10^{-2}$ respectively for curriculum reinforcement learning. For the number of specialists, we set $e = 3$. The details are shown in Appendix F.2.

**Dataset Splitting and Difficulty Stratification.** We adhere strictly to the official guidelines for partitioning the training, validation, and test sets. To facilitate our curriculum learning strategy, we further categorize samples into three difficulty levels (*Easy*, *Medium*, *Hard*) based on the consistency and accuracy of specialist responses. It is important to note that this difficulty grading is utilized exclusively during the training phase to schedule the curriculum. During inference, the model generates responses without access to difficulty labels or ground truth information. The difficulty-based breakdown presented in our analysis is applied *post-hoc* to the test set solely to visualize model robustness against specialist noise, ensuring no data leakage.

**Baseline Methods.** We compare MMedAgent-RL with methods under two different settings: 1) Single-agent setting: This includes a series of state-of-the-art LVLMs, encompassing both general LVLMs and domain-specific LVLMs. Specifically, we include comparisons of the LLaVA series (Liu et al., 2024a), Yi-VL-34B (Young et al., 2024), Qwen-VL (Bai et al., 2025), LLaVA-Med (Li et al., 2023), MedFlamingo (Moor et al., 2023), RadFM (Wu et al., 2023), MedVLThinker-7B Huang et al. (2025) and GPT-4o (OpenAI, 2024). 2) Multi-agent setting: This includes MedAgents (Tang et al., 2024), MDAgents (Kim et al., 2024), AFlow (Zhang et al., 2024).

**Data and Metrics.** We train on the three medical VQA datasets, i.e., VQA-RAD (Lau et al., 2018), SLAKE (Liu et al., 2021), PathVQA (He et al., 2020). Their test sets are considered the in-domain test sets. Additionally, following Chen et al. (2024a), we select the health and medicine subset of MMMU (Yue et al., 2024), and OmniMedVQA (Hu et al., 2024b) as out-of-distribution datasets. All evaluation questions are multiple-choice, and accuracy is used as the evaluation metric.

## 5.2 MAIN RESULTS

In this section, we conduct a comprehensive comparison on the medical VQA task involving six datasets and various LVLMs as well as baseline methods based on Qwen2.5-VL.

**Comparison with Baselines in In-Distribution Datasets.** Table 1 shows the performance of various models across four medical VQA benchmarks. General LVLMs like LLaVA-v1.6-34B and GPT-4o exhibit consistently strong performance, outperforming earlier medical-specific models such as Med-Flamingo and RadFM. Notably, GPT-4o achieves the highest average score (68.6%) among all single-agent models, demonstrating its powerful generalization capabilities even in specialized medical domains. Interestingly, the multi-agent collaboration strategy further boosts performance. MMedAgent-RL achieves the best overall average (73.3%), surpassing even the strongest single-agent models. This highlights the effectiveness of collaborative inference in leveraging the complementary strengths of different models. In addition, we utilized majority voting as the method for test-time scaling (Snell et al., 2024), which further improved the model performance by 4.5%. This demonstrates that optimizing token entropy during the RL process is positive for efficient sampling. For different specialists, a varied spirit of exploration remains very important.

**Performance in Out-of-Distribution Datasets.** We evaluate the performance of MMedAgent-RL across various out-of-distribution (OOD) datasets. The results are presented in Table 1, which demonstrates the generalization of our approach in adapting to different OOD scenarios. These two OOD datasets cover multiple body parts and involve various medical image modalities. Through reinforcement learning, MMedAgent-RL demonstrates significant superiority across multiple modalities, outperforming the base model by 21% and the SFT method by 23.6%. Moreover, it surpasses the performance of multi-agent collaboration methods that cannot optimize models, i.e., MedAgents, MDAgents and AFlow, by 23%, 19% and 17%, highlighting the effectiveness of our approach in handling diverse and unseen data distributions.

## 5.3 ANALYSIS

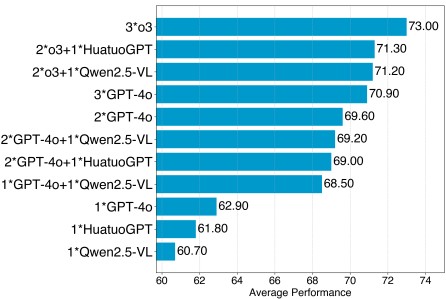

Figure 3: Results of different settings of specialist doctors.

Figure 4: Results under different levels of decision difficulty.

In this section, we conduct a detailed performance analysis at each step and explore how model type, numbers of specialist doctors, and varying levels of decision difficulty affect the results, to better understand the performance gains achieved by MMedAgent-RL.

**Ablation Studies.** We conducted a series of ablation experiments to evaluate the impact of each component in MMedAgent-RL, as shown in Table 2. We can see that: (1) Reliable triage doctors are important. Accurately determining the department to which a specialist doctor belongs helps the model call upon knowledge from their corresponding field of expertise to answer questions, improving the accuracy of specialist doctors' answers. A fine-tuned triage doctor significantly improves model performance compared to the original model, with an average performance increase of 3% across multiple datasets. (2) Based on this, the mechanism of specialist doctor consultation is introduced, further helping the decision-making agent fully utilize expert opinions, with an average performance increase of 4.5% across multiple datasets. (3) Most importantly, the addition of curriculum multi-agent reinforcement learning (C-MARL) enhanced the decision-making agent's understanding of specialist doctors' knowledge, achieving a significant performance improvement of 18.6%. This indicates that C-MARL can effectively solve the problem of overall misalignment between the model and external

knwoledge. Specifically, each stage plays a corresponding role and can understand the specialist doctors' knowledge according to the goals of different stages, achieving overall performance gains.

**Analysis of Specialist Doctors.** We analyze the types and number of models playing the role of specialist doctors. Specifically, as shown in Figure 3, regarding the model types, the performance of the final decision-making agent is closely related to the performance of the specialist doctors. Therefore, we used a series of models that performed well on multiple datasets, such as o3, GPT-4o, HuatuoGPT-Vision, and Qwen2.5-VL, as specialist doctors. Since

Table 2: Ablation results on ID and OOD datasets.

| Model | ID | | OOD | |
|---|---|---|---|---|
| | VQA-RAD | SLAKE | OmniMedVQA | MMMU |
| MMedAgent-RL | 71.5 | 76.2 | 73.3 | 71.9 |
| w/o Triage | 66.3 | 69.9 | 66.2 | 59.3 |
| w/o Specialists | 65.8 | 67.8 | 64.4 | 54.2 |
| w/o C-MARL | 63.5 | 65.5 | 57.9 | 50.2 |
| + Easy | 64.7 | 69.3 | 68.2 | 58.0 |
| + Medium | 69.4 | 76.9 | 70.8 | 68.8 |
| + Hard | 71.5 | 76.2 | 73.3 | 71.9 |

the areas or tasks that each model excels in are not completely consistent, the specialist doctor played by o3 ultimately performed the best. Its performance across various aspects was relatively balanced, enabling MMedAgent-RL to achieve the best performance. Refer to Appendix G.2 for details.

**Performance under Different Levels of Decision Difficulty.** To analyze the robustness of the GP agent against noisy advice, we categorize test samples into three difficulty levels based on the consistency and correctness of specialist outputs: *Easy* (all specialists are correct), *Medium* (specialists disagree), and *Hard* (specialists consistently provide incorrect advice, creating a misleading consensus). Figure 4 illustrates the performance of baselines versus MMedAgent-RL across these categories. While specialist noise—particularly in the 'Hard' setting—significantly hampers the baseline model's decision-making, our C-MARL method enables the agent to gradually learn to distinguish and selectively utilize specialist knowledge. Consequently, MMedAgent-RL achieves an overall performance improvement of 20%, with the most significant gains observed in overcoming misleading specialist hallucinations in the hard cases.

**Case Study and Outlook on "Aha Moments".** As shown in Figure 5, MMedAgent-RL demonstrates strong performance across multiple cases. It provides accurate answers within the `<answer>` tags and generates high-quality reasoning resembling that of human doctors: defining the disease, analyzing images, and checking consistency with the definition. It also evaluates specialists' outputs before reasoning out the correct answer. While lacking the "aha moment" observed in humans, this structured reasoning highlights the potential for more human-like scientific AI systems.

# 6 RELATED WORK

**Medical Vision-Language Models.** The advancement of Vision-Language Models (VLMs) (Liu et al., 2024a;b; Zhu et al., 2023; Bai et al., 2023; Chen et al., 2024c) has catalyzed significant progress in medical applications (Xia et al., 2024a;c;b; Chen et al., 2024a; Zhu et al., 2024), with large-scale models like LLaVA-Med (Li et al., 2024b), HuatuoGPT-Vision (Chen et al., 2024a), and VILA-M3 (Nath et al., 2024) demonstrating profound results in medical diagnostics. However, single models struggle to handle cross-domain expertise. Although multi-agent systems (Li et al., 2024c; Kim et al., 2024; Tang et al., 2024; Zhang et al., 2024) have been proposed to combine diverse medical expertise, existing approaches typically use preset workflows that lack adaptive reasoning capabilities. Works such as MedAgentsBench (Tang et al., 2025) and AI Hospital (Fan et al., 2025) have established frameworks for evaluating multi-agent interactions and diagnostic thinking. Furthermore, AgentClinic (Schmidgall et al., 2024) introduces a multimodal benchmark to simulate clinical environments specifically for evaluating embodied agents. Additionally, most Med-VLMs are developed through supervised fine-tuning (SFT) on general VLMs using biomedical instruction data (Chen et al., 2024a; Li et al., 2023; Liu et al., 2023), which is limited by the scarcity of high-quality reasoning examples and often results in models that struggle with complex diagnostic reasoning across specialties.

**Reinforcement Learning for Multimodal Reasoning.** To address the limitations of static multi-agent systems and overcome the constraints of supervised fine-tuning, Reinforcement Learning (RL) offers a promising alternative for optimizing medical reasoning. RL evolves from establishing foundational frameworks for learning from human preferences (Christiano et al., 2017; Ziegler et al., 2019) to developing sophisticated approaches like RLHF for instruction following (Ouyang et al., 2022) and self-correction (Kumar et al., 2024; Wu et al., 2025). Recent advances with DeepSeek-R1 (Guo et al., 2025) demonstrate that LLMs can leverage RL to enhance reasoning capabilities in

**Question**: does muscle show kidney, adult polycystic kidney? 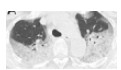
**Options**: A. yes, B. no.
**Answer**: B. no.

**Ours**: To determine if muscle shows kidney or adult polycystic kidney, we would typically look for specific characteristics such as the presence of renal tubules, glomeruli, and the typical appearance of kidney tissue under a microscope. The radiologist's report mentions "no," which indicates that this image is not related to adult polycystic kidney disease. The pathologist's report mentions "Not". Given these points: it is unlikely that muscle shows kidney or adult polycystic kidney. <answer>B. no</answer>

**Question**: does the histologic appearance demonstrate small round, uniform nuclei and moderate cytoplasm? 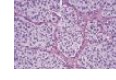
**Options**: A. yes, B. no.
**Answer**: A. yes.

**Ours**: The histologic appearance described in the image shows small round, uniform nuclei and moderate cytoplasm, which is characteristic of certain types of cancer cells, such as those found in breast or ovarian cancer. The pathologist's suggestion of "No, the nuclei vary in size" is less likely because the nuclei in the image are clearly of uniform size, which usually doesn't show these characteristics. <answer>A. yes</answer>

Figure 5: Several case analyses. In the model responses, blue text represents the process of reasoning about relevant medical knowledge based on the question, and red text represents the analysis of the answer provided by the specialists.

complex tasks without supervision, showing exceptional performance in mathematics and coding challenges (Yeo et al., 2025). This success has extended to multimodal reasoning (Meng et al., 2025; Shen et al., 2025; Wang et al., 2025a; Liu et al., 2025b; Geng et al., 2025; Zhang et al., 2025b; Xia et al., 2025; Liu et al., 2025a; Su et al., 2025), including the biomedical domain (Pan et al., 2025). DoctorAgent-RL (Feng et al., 2025) employs a multi-agent collaborative RL system specifically to optimize multi-turn clinical dialogue policies. Then Med-U1 (Zhang et al., 2025a) utilizes large-scale RL to incentivize unified reasoning patterns in medical LLMs. However, prior RL-based approaches for multimodal reasoning have primarily focused on optimizing a single model, leaving the potential of RL for enhancing multi-agent medical collaboration largely unexplored.

# 7 CONCLUSION

This work presents MMedAgent-RL, a novel RL framework for multi-agent collaboration in medical multimodal reasoning. The framework mimics a clinical "triage-and-referral" system, using a curriculum RL strategy to train a primary model to intelligently handle noisy or conflicting inputs from different "specialist" agents. Experiments demonstrate the method's strong performance across multiple medical visual question answering datasets, offering a promising new direction for building reasoning models that more closely emulate human diagnostic thinking.

**Ethics Statement**. All authors have read and comply with the ICLR Code of Ethics. This work does not involve human subjects or sensitive data, and we are unaware of any potential misuse, harm, or bias. No conflicts of interest or compromising sponsorships exist.

**Reproducibility Statement** Details of the proposed methodology, training procedure, hyperparameters, and evaluation metrics are provided in Section F. We include complete algorithm pseudocode in Alg. 1 and a full description of the datasets in Appendix D and Appendix F.1.

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

APPENDIX

## A   LARGE LANGUAGE MODEL USAGE

All content in this article is entirely authored by the writers. The LLM (we use Gemini2.5-Pro) was used solely for language refinement and stylistic polishing, without contributing to content generation. All LLM-refined passages were subsequently reviewed and revised by the authors.

## B   OMITTED THEOREMS AND PROOFS

### B.1   NOTATION

We denote by $\mathcal{R}_n(\mathcal{F}) := \mathbb{E}_{S=(x_1,\ldots,x_n)}\big[\mathbb{E}_\sigma\big[\sup_{f\in\mathcal{F}}\frac{1}{n}\sum_{i=1}^n \sigma_i f(x_i)\big]\big]$ the (expected) Rademacher complexity of the function class $\mathcal{F} \subseteq \{f : \mathcal{X} \to \mathbb{R}\}$, where $\sigma_1,\ldots,\sigma_n$ are i.i.d. Rademacher variables taking values in $\{-1, +1\}$ with probability $1/2$ each and the outer expectation is over the sample $S$. The Kullback–Leibler (KL) divergence from a discrete distribution $p$ to a discrete distribution $q$ (defined over a common support $\mathcal{X}$) is given by $\mathbb{D}_{\mathrm{KL}}[p\,\|\,q] := \sum_{x\in\mathcal{X}} p(x)\log\left(\frac{p(x)}{q(x)}\right)$, under the assumption that whenever $p(x) > 0$, one also has $q(x) > 0$ for all $x \in \mathcal{X}$. For two positive sequences $\{a_n\}$ and $\{b_n\}$, write $a_n = o(b_n)$ if $\lim_n a_n/b_n = 0$, $a_n = O(b_n)$ if $a_n \leq Cb_n$, and $a_n = \Omega(b_n)$ if $a_n \geq Cb_n$ for all $n$ and some positive $C$. The $l_2$ norm of a vector $x \in \mathbb{R}^d$ is defined as $\|x\|_2 := \left(\sum_{i=1}^d x_i^2\right)^{1/2}$. For a measurable function $f : \mathcal{X} \to \mathbb{R}$, the $L_\infty$ norm is $\|f\|_\infty := \sup_{x\in\mathcal{X}} |f(x)|$, which equals $\sup_{x\in\mathcal{X}} |f(x)|$ when $\mathcal{X}$ is finite.

### B.2   SOME USEFUL LEMMAS

**Lemma B.1** (Theorem 4.10 in Wainwright (2019)). *For any $b$-uniformly bounded class of functions $\mathcal{F}$, any positive integer $n \geq 1$, and any scalar $\delta \geq 0$, we have*

$$\mathbb{P}\left(\sup_{f\in\mathcal{F}} |(\mathbb{P}_n - \mathbb{P})[f]| \leq 2\mathcal{R}_n(\mathcal{F}) + \delta\right) \geq 1 - \exp\left(-\frac{n\delta^2}{2b^2}\right).$$

*A function class $\mathcal{F}$ is said to be $b$-uniformly bounded if $\|f\|_\infty \leq b, \forall f \in \mathcal{F}$.*

To prove Theorem 4.1, we split the analysis into two parts: Theorem B.1, which establishes the upper bound for curriculum learning, and Theorem B.2, which establishes the lower bound for standard SGD. Combining these results yields Theorem 4.1.

### B.3   ANALYSIS FOR CURRICULUM LEARNING

In this section, we systematically develop the theoretical foundation for curriculum learning. As previously discussed, we reiterate here that the curriculum learning strategy is effective for policy learning in the reinforcement learning (RL) setting. Our goal is to quantify the number of samples required to achieve a target error level $\epsilon$.

Recall the distribution assumption:

**Assumption B.1.** *From the easiest to the hardest dataset, for $j$-th dataset, we assume*

$$\mathcal{D}_j = \left\{(x_i, y_i)_{i=1}^{n_j} \mid x_i \sim p(x),\ y_i \sim \pi_{\theta_j^*}(y \mid x)\right\}.$$

Recall that the GRPO loss is given by

$$\mathcal{J}_{\mathrm{GRPO}}(\theta) = \mathbb{E}_{x,\{y_i\}}\left[\frac{1}{G}\sum_{i=1}^G \left(\min\left(r_i A_i, \mathrm{clip}(r_i, 1-\epsilon, 1+\epsilon)A_i\right) - \beta\,\mathbb{D}_{\mathrm{KL}}(\pi_\theta \| \pi_{\mathrm{ref}})\right)\right], r_i = \frac{\pi_\theta(y_i \mid x)}{\pi_{\mathrm{old}}(y_i \mid x)}.$$
(B.1)

And our loss function is defined as

$$\mathcal{J}_{\text{C-MARL}}(\theta) = \mathbb{E}\left[\mathcal{J}_{\mathrm{GRPO}}(\theta) + \gamma_s \cdot H_t(\pi_{\theta_{\mathrm{GP}}^{\mathrm{attend}}})\right], \quad H_t = -\sum_{j=1}^V p_{t,j}\log p_{t,j},$$
(B.2)

$$\text{where} \quad \mathbf{p}_t = \pi_\theta(\cdot \mid \mathcal{R}_{<t}, x; T) = \mathrm{Softmax}\left(\frac{\mathbf{z}_t}{\tau}\right).$$

When minimizing the objective, we can interpret the procedure as performing SGD on the batch loss. Specifically, define

$$\mathcal{L}_j(\theta) = -\mathbb{E}_{x \sim p(x),\, y \sim \pi_{\theta_j^*}(y|x)}\Big[\mathcal{J}_{\mathrm{GRPO}}(\theta) + \gamma_s \cdot H_t(\pi_{\theta_{\mathrm{GP}}^{\mathrm{attend}}})\Big].$$

It is natural to introduce the following self-consistency condition:

**Assumption B.2** (Self-consistency). *$\theta_j^*$ is the minimizer of $\mathcal{L}_j(\theta)$.*

Under this assumption, minimizing $\mathcal{L}_j(\theta)$ allows us to recover the true policy.

For the empirical version, the dataset at stage $j$ is

$$\mathcal{D}_j = \Big\{ (x_i, y_i)_{i=1}^{n_j} \mid x_i \sim p(x),\ y_i \sim \pi_{\theta_j^*}(y \mid x) \Big\},$$

and the overall dataset is

$$\mathcal{D} = \bigcup_{j=1}^{J} \mathcal{D}_j.$$

The corresponding empirical approximation of the batch loss is

$$\tilde{\mathcal{L}}_j(\theta) = -\frac{1}{n_j} \sum_{(x,y) \in \mathcal{D}_j} \Big[\mathcal{J}_{\mathrm{GRPO}}(\theta) + \gamma_s \cdot H_t(\pi_{\theta_{\mathrm{GP}}^{\mathrm{attend}}})\Big].$$

We denote its minimizer by $\tilde{\theta}_j$.

The overall training procedure then consists of successively minimizing $\tilde{\mathcal{L}}_j(\theta)$ for $j = 1, \ldots, J$, where at each stage $j$ we perform $K_j$ iterations of SGD.

We first apply classical learning theory techniques to establish the convergence between $\tilde{\mathcal{L}}_j(\theta)$ and $\mathcal{L}_j(\theta)$. After that, we analyze the behavior of the SGD iterations.

To proceed, we introduce two assumptions on the loss function class

$$\mathcal{F} = \Big\{ \mathcal{L}_j(\theta),\ \tilde{\mathcal{L}}_j(\theta) \mid j = 1, \ldots, J \Big\}.$$

**Assumption B.3** (Boundedness). *For any loss function $f \in \mathcal{F}$, $\|f\|_\infty \leq B_1$.*

**Assumption B.4.** *For any loss function $f \in \mathcal{F}$, let $\theta_f^*$ denote its minimizer. If $f(\theta) - f(\theta_f^*) \leq U_1$, then*

$$L_1\big\|\theta - \theta_f^*\big\|_2^2 \leq f(\theta) - f(\theta_f^*).$$

*Remark* B.1. Here we assume that, in a neighborhood of its minimizer, the objective is locally convex. Our GRPO loss satisfies this assumption.

**Proposition B.1.** *Suppose Assumptions B.1–B.4 hold. At the $j$-th step, for any $0 < \epsilon_1 < U_1$, assume the sample size $n_j$ is sufficiently large so that*

$$\mathcal{R}_{n_j}(\Pi) \leq \frac{\epsilon_1}{4},$$

*where $\Pi = \{\pi_\theta(y \mid x) : \theta \in \Theta\}$ denotes the distribution space induced by the parameter space $\Theta$. Define the event*

$$\Omega_j^{(1)} = \left\{ \big\|\tilde{\theta}_j - \theta_j^\star\big\|_2^2 \leq \frac{\epsilon_1}{L_1} \right\}.$$

*Then,*

$$\mathbb{P}\big(\Omega_j^{(1)}\big) \geq 1 - \exp\left(-\frac{n_j \epsilon_1^2}{2B_1^2}\right).$$

*In other words, with high probability, the empirical minimizer $\tilde{\theta}_j$ lies close to the population minimizer $\theta_j^\star$ in the parameter space, and the estimation error decreases as the sample size $n_j$ increases.*

*Remark* B.2. Usually, for regular parameter space, $\mathcal{R}_{n_j}(\Pi)$ is $o(1)$.

*Proof.* We begin by decomposing the excess risk:

$$\mathcal{L}_j(\tilde{\theta}_j) - \mathcal{L}_j(\theta_j^\star) \leq \mathcal{L}_j(\tilde{\theta}_j) - \tilde{\mathcal{L}}_j(\tilde{\theta}_j) + \tilde{\mathcal{L}}_j(\tilde{\theta}_j) - \tilde{\mathcal{L}}_j(\theta_j^\star) + \tilde{\mathcal{L}}_j(\theta_j^\star) - \mathcal{L}_j(\theta_j^\star).$$

Since $\tilde{\theta}_j$ minimizes $\tilde{\mathcal{L}}_j(\theta)$, the middle term is non-positive. Hence,

$$\mathcal{L}_j(\tilde{\theta}_j) - \mathcal{L}_j(\theta_j^\star) \leq 2 \sup_{\theta \in \Theta} \left| \tilde{\mathcal{L}}_j(\theta) - \mathcal{L}_j(\theta) \right|.$$

Applying Lemma B.1, we obtain the high-probability bound

$$\mathbb{P}\Big( \mathcal{L}_j(\tilde{\theta}_j) - \mathcal{L}_j(\theta_j^\star) \leq 2\mathcal{R}_{n_j}(\Pi) + \frac{\epsilon_1}{2} \Big) \geq 1 - \exp\Big( -\frac{n_j \epsilon_1^2}{2B_1^2} \Big).$$

If $n_j$ is chosen sufficiently large such that

$$2\mathcal{R}_{n_j}(\Pi) \leq \frac{\epsilon_1}{2},$$

then with the same probability we have

$$\mathcal{L}_j(\tilde{\theta}_j) - \mathcal{L}_j(\theta_j^\star) \leq \epsilon_1.$$

Moreover, if $\epsilon_1 \leq U_1$, Assumption B.4 implies

$$L_1 \left\| \tilde{\theta}_j - \theta_j^\star \right\|_2^2 \leq \mathcal{L}_j(\tilde{\theta}_j) - \mathcal{L}_j(\theta_j^\star).$$

Combining the above, we obtain

$$\mathbb{P}\left( \left\| \tilde{\theta}_j - \theta_j^\star \right\|_2^2 \leq \frac{\epsilon_1}{L_1} \right) = \mathbb{P}\Big( \Omega_j^{(1)} \Big) \geq 1 - \exp\Big( -\frac{n_j \epsilon_1^2}{2B_1^2} \Big).$$

This establishes the claim. $\qquad\square$

To establish the iteration complexity at stage $j$, we impose two standard conditions on the loss function class $\mathcal{F}$.

**Assumption B.5** (Smoothness)**.** *Let $L_2 > 0$. For any loss function $f \in \mathcal{F}$, the gradient of $f$ is $L_2$-Lipschitz continuous. That is, for all $\theta, \tilde{\theta} \in \Theta$,*

$$\left\| \nabla f(\theta) - \nabla f(\tilde{\theta}) \right\|_2 \leq L_2 \left\| \theta - \tilde{\theta} \right\|_2.$$

**Assumption B.6** (Polyak–Łojasiewicz (PL) condition)**.** *For any loss function $f \in \mathcal{F}$, we assume that $f$ satisfies the PL inequality with parameter $\mu > 0$. Specifically, for all $\theta \in \Theta$,*

$$f(\theta) - f(\theta_f^\star) \leq \frac{1}{2\mu} \|\nabla f(\theta)\|_2^2,$$

*where $\theta_f^\star = \arg\min_{\theta \in \Theta} f(\theta)$ denotes the minimizer of $f$.*

Based on Assumptions B.5 and B.6, Lei et al. (2019) established the following result.

**Lemma B.2.** *Suppose Assumptions B.5 and B.6 hold, and that $\nabla f(\theta_f^\star) = 0$. If the step size satisfies $\eta_t = \eta \leq \mu/L^2$, then*

$$\mathbb{E}[f(\theta_{t+1})] - f(\theta_f^\star) \leq (1 - \mu\eta)^t \big( f(\theta_1) - f(\theta_f^\star) \big),$$

*where $f \in \mathcal{F}$ and $\theta_f^\star = \arg\min_{\theta \in \Theta} f(\theta)$.*

Next proposition give the detail how many iteration do we need in $j$ step SGD

**Proposition B.2.** *Suppose Assumptions B.1, B.5, B.6, and B.4 hold. Fix stage $j$. Run SGD on the empirical loss $\tilde{\mathcal{L}}_j$ with a constant stepsize $\eta \leq \mu/L_2^2$ for $K_j$ iterations, starting from $\widehat{\theta}_{j,0} = \widehat{\theta}_{j-1,K_{j-1}}$. Let*

$$D_{j-1} = \left\| \widehat{\theta}_{j,0} - \tilde{\theta}_j \right\|_2^2.$$

*Define the event*

$$\Omega_j^{(2)} = \left\{ \left\| \widehat{\theta}_{j,K_j} - \tilde{\theta}_j \right\|_2^2 \leq \frac{\epsilon_1}{L_1} \right\}.$$

*Then for any $0 < \epsilon_1 \leq U_1$, if*

$$K_j = O\left( \frac{1}{\mu\eta} \log\left( \frac{L_2^2 D_{j-1}}{2\mu \epsilon_1^2} \right) \right),$$

*we have*

$$\mathbb{P}\left( \Omega_j^{(2)} \right) \geq 1 - \epsilon_1.$$

*Proof.* Firstly, applying Lemma B.2 (with $f = \tilde{\mathcal{L}}_j$ and $\eta_t = \eta \leq \mu/L^2$), we obtain for $K_j \geq 0$,

$$\mathbb{E}\left[ \tilde{\mathcal{L}}_j(\widehat{\theta}_{j,K_j}) - \tilde{\mathcal{L}}_j(\tilde{\theta}_j) \right] \leq (1 - \mu\eta)^{K_j} \left( \tilde{\mathcal{L}}_j(\widehat{\theta}_{j,0}) - \tilde{\mathcal{L}}_j(\tilde{\theta}_j) \right).$$

By Assumptions B.6 (PL) and B.5 (Lipschitz gradient), we have

$$\tilde{\mathcal{L}}_j(\widehat{\theta}_{j,0}) - \tilde{\mathcal{L}}_j(\tilde{\theta}_j) \leq \frac{1}{2\mu} \left\| \nabla \tilde{\mathcal{L}}_j(\widehat{\theta}_{j,0}) \right\|_2^2 \leq \frac{L_2^2}{2\mu} \left\| \widehat{\theta}_{j,0} - \tilde{\theta}_j \right\|_2^2.$$

Taking expectations over the randomness up to stage $j-1$ (recall $\widehat{\theta}_{j,0} = \widehat{\theta}_{j-1,K_{j-1}}$), we get

$$\mathbb{E}\left[ \tilde{\mathcal{L}}_j(\widehat{\theta}_{j,K_j}) - \tilde{\mathcal{L}}_j(\tilde{\theta}_j) \right] \leq \frac{L_2^2}{2\mu} (1 - \mu\eta)^{K_j} \mathbb{E}\left[ \left\| \widehat{\theta}_{j,0} - \tilde{\theta}_j \right\|_2^2 \right] = \frac{L_2^2}{2\mu} (1 - \mu\eta)^{K_j} D_{j-1}.$$

Choose $K_j$ so that

$$\frac{L_2^2}{2\mu} (1 - \mu\eta)^{K_j} D_{j-1} \leq \epsilon_1^2.$$

Using $1 - \mu\eta \leq e^{-\mu\eta}$, a sufficient condition is

$$K_j \geq \frac{1}{\mu\eta} \log\left( \frac{L_2^2 D_{j-1}}{2\mu \epsilon_1} \right).$$

By Markov's inequality, with probability at least $1 - \epsilon_1$,

$$\Omega_j^{(2)} = \left\{ \left| \tilde{\mathcal{L}}_j(\widehat{\theta}_{j,K_j}) - \tilde{\mathcal{L}}_j(\tilde{\theta}_j) \right| \leq \epsilon_1 \right\}$$

occurs. On $\Omega_j^{(2)}$, Assumption B.4 yields

$$\left\| \widehat{\theta}_{j,K_j} - \tilde{\theta}_j \right\|_2^2 \leq \frac{1}{L_1} \left| \tilde{\mathcal{L}}_j(\widehat{\theta}_{j,K_j}) - \tilde{\mathcal{L}}_j(\tilde{\theta}_j) \right| \leq \frac{\epsilon_1}{L_1},$$

which completes the proof. $\qquad\square$

**Theorem B.1.** *Suppose Assumptions B.1–B.6 hold. Fix $0 < \epsilon_1 \leq U_1$. For each stage $j = 1, \ldots, J$ choose $n_j$ such that $\mathcal{R}_{n_j}(\Theta) \leq \epsilon_1/4$, and run SGD with constant stepsize $\eta \leq \mu/L_2^2$. A sufficient total number of iterations is*

$$K = O\left( \frac{1}{\mu\eta} \sum_{j=0}^{J-1} \log \frac{L_2^2 \|\theta_j^\star - \theta_{j+1}^\star\|_2^2}{\mu \epsilon_1} \right).$$

*With this $K$, the final iterate satisfies $\|\widehat{\theta}_{J,K} - \theta_J^\star\|_2^2 \leq \epsilon_1/L_1$ with probability at least $1 - \sum_{j=1}^J \exp\left( -\frac{n_j \epsilon_1^2}{2B_1^2} \right) - J\epsilon_1$.*

*Proof.* Recall the following events from Propositions B.1 and B.2:

$$\Omega_j^{(1)} = \left\{ \left\| \tilde{\theta}_j - \theta_j^\star \right\|_2^2 \leq \tfrac{\epsilon_1}{L_1} \right\}, \qquad \Omega_j^{(2)} = \left\{ \left\| \widehat{\theta}_{j,K_j} - \tilde{\theta}_j \right\|_2^2 \leq \tfrac{\epsilon_1}{L_1} \right\}.$$

By Proposition B.1 and the choice of $n_j$,

$$\mathbb{P}\left( \Omega_j^{(1)} \right) \geq 1 - \exp\left( -\frac{n_j \epsilon_1^2}{2B_1^2} \right).$$

By Proposition B.2 and the choice of $K_j$,

$$\mathbb{P}\left( \Omega_j^{(2)} \right) \geq 1 - \epsilon_1.$$

Let

$$\Omega = \bigcap_{j=1}^{J} \left( \Omega_j^{(1)} \cap \Omega_j^{(2)} \right).$$

A union bound gives

$$\mathbb{P}(\Omega) \geq 1 - \sum_{j=1}^{J} \exp\left( -\frac{n_j \epsilon_1^2}{2B_1^2} \right) - J\epsilon_1.$$

Condition on $\Omega$. From the end of stage $j$ to the start of stage $j+1$ we have $\widehat{\theta}_{j,K_j} = \widehat{\theta}_{j+1,0}$, and thus

$$\left\| \widehat{\theta}_{j+1,0} - \tilde{\theta}_{j+1} \right\|_2 \leq \left\| \widehat{\theta}_{j,K_j} - \tilde{\theta}_j \right\|_2 + \left\| \tilde{\theta}_j - \theta_j^\star \right\|_2 + \left\| \theta_j^\star - \theta_{j+1}^\star \right\|_2 + \left\| \theta_{j+1}^\star - \tilde{\theta}_{j+1} \right\|_2$$

$$\leq \sqrt{\tfrac{\epsilon_1}{L_1}} + \sqrt{\tfrac{\epsilon_1}{L_1}} + \left\| \theta_j^\star - \theta_{j+1}^\star \right\|_2 + \sqrt{\tfrac{\epsilon_1}{L_1}} = 3\sqrt{\tfrac{\epsilon_1}{L_1}} + \left\| \theta_j^\star - \theta_{j+1}^\star \right\|_2,$$

where we used $\Omega_j^{(2)}$ and Assumption B.4 to bound $\left\| \widehat{\theta}_{j,K_j} - \tilde{\theta}_j \right\|_2^2 \leq \epsilon_1/L_1$, and $\Omega_j^{(1)}, \Omega_{j+1}^{(1)}$ to bound the two terms involving $\tilde{\theta}_j$ and $\tilde{\theta}_{j+1}$. Squaring both sides and using $(a+b)^2 \leq 2a^2 + 2b^2$ gives the displayed recursion

$$D_j = \left\| \widehat{\theta}_{j+1,0} - \tilde{\theta}_{j+1} \right\|_2^2 \leq \frac{18\,\epsilon_1}{L_1} + 2\left\| \theta_j^\star - \theta_{j+1}^\star \right\|_2^2.$$

By Proposition B.2, choosing

$$K_j = O\left( \frac{1}{\mu\eta} \log\left( \frac{L_2^2 \left\| \theta_{j-1}^\star - \theta_j^\star \right\|_2^2}{\mu\,\epsilon_1} \right) \right)$$

ensures $\Omega_j^{(2)}$ holds and

$$\left\| \widehat{\theta}_{j,K_j} - \tilde{\theta}_j \right\|_2^2 \leq \frac{\epsilon_1}{L_1}.$$

Iterating this from $j = 1$ to $J$, we obtain at the final stage

$$\left\| \widehat{\theta}_{J,K_J} - \theta_J^\star \right\|_2 \leq \left\| \widehat{\theta}_{J,K_J} - \tilde{\theta}_J \right\|_2 + \left\| \tilde{\theta}_J - \theta_J^\star \right\|_2 \leq \sqrt{\tfrac{\epsilon_1}{L_1}} + \sqrt{\tfrac{\epsilon_1}{L_1}},$$

and hence

$$\left\| \widehat{\theta}_{J,K_J} - \theta_J^\star \right\|_2^2 \leq \frac{4\epsilon_1}{L_1}.$$

Finally, the total number of iterations is $K = \sum_{j=1}^{J} K_j$, and the total sample size is $n = \sum_{j=1}^{J} n_j$. $\qquad \square$

### B.4 Analysis for Standard SGD

Without loss of generality, assume $n_1 = \cdots = n_J = n_{\mathrm{rg}}/J$, where $n_{\mathrm{rg}}$ is the total sample size, and suppose we run SGD directly on the dataset $\mathcal{D}$. Under this equal-allocation setting, the pooled data are drawn from the uniform mixture

$$\pi_{\theta^\star_{\mathrm{rg}}}(y \mid x) = \frac{1}{J} \sum_{j=1}^{J} \pi_{\theta^\star_j}(y \mid x).$$

Define the mixed (population) and empirical losses by

$$\mathcal{L}_{\mathrm{rg}}(\theta) = -\mathbb{E}_{x \sim p(x),\, y \sim \pi_{\theta^\star_{\mathrm{rg}}}(y|x)} \Big[ \mathcal{J}_{\mathrm{GRPO}}(\theta) + \gamma_s H_t(\pi_{\theta^{\mathrm{attend}}_{\mathrm{GP}}}) \Big],$$

$$\tilde{\mathcal{L}}_{\mathrm{rg}}(\theta) = -\frac{1}{n_{\mathrm{rg}}} \sum_{(x,y) \in \mathcal{D}} \Big[ \mathcal{J}_{\mathrm{GRPO}}(\theta) + \gamma_s H_t(\pi_{\theta^{\mathrm{attend}}_{\mathrm{GP}}}) \Big].$$

Let $\theta^\star_{\mathrm{rg}}$ and $\tilde{\theta}_{\mathrm{rg}}$ denote the minimizers of $\mathcal{L}_{\mathrm{rg}}$ and $\tilde{\mathcal{L}}_{\mathrm{rg}}$, respectively.

$$\delta = \|\theta^\star_{\mathrm{rg}} - \theta^\star\|_2.$$

**Theorem B.2.** *If $n_{\mathrm{rg}}$ satisfies $\mathcal{R}_{n_{\mathrm{rg}}}(\Pi) \leq \epsilon_1/4$ and SGD uses the same stepsize $\eta$, take*

$$K_{\mathrm{rg}} = O\left( \frac{1}{\mu\eta} \log \frac{L_2^2 \|\theta^\star_0 - \theta^\star_{\mathrm{rg}}\|_2^2}{\mu\,\epsilon_1} \right).$$

*Then, with probability at least $1 - \exp\Big( - \frac{n_{\mathrm{rg}}\epsilon_1^2}{2B_1^2} \Big) - \epsilon_1$, we have*

$$\|\widehat{\theta}_{\mathrm{rg}} - \theta^\star\|_2 \geq \frac{\delta}{2} \quad \text{whenever} \quad \epsilon_1 \leq \frac{L_1\delta^2}{16}.$$

*Proof.* Apply Proposition B.1 to $\tilde{\mathcal{L}}_{\mathrm{rg}}$: with probability at least $1 - \exp\big( -(n_{\mathrm{rg}}\epsilon_1^2)/(2B_1^2) \big)$,

$$\|\tilde{\theta}_{\mathrm{rg}} - \theta^\star_{\mathrm{rg}}\|_2^2 \leq \frac{\epsilon_1}{L_1}.$$

Next apply Proposition B.2 to the standard (pooled) SGD (same $\eta \leq \mu/L_2^2$) and $K_{\mathrm{rg}} = O\Big( \frac{1}{\mu\eta} \log \frac{L_2^2 \|\theta^\star_0 - \theta^\star_{\mathrm{rg}}\|_2^2}{\mu\,\epsilon_1} \Big)$: with probability at least $1 - \epsilon_1$,

$$\|\widehat{\theta}_{\mathrm{rg}} - \tilde{\theta}_{\mathrm{rg}}\|_2^2 \leq \frac{\epsilon_1}{L_1}.$$

By a union bound, both events hold simultaneously with probability at least $1 - \exp\Big( - \frac{n_{\mathrm{rg}}\epsilon_1^2}{2B_1^2} \Big) - \epsilon_1$.

On this event, the triangle inequality yields

$$\|\widehat{\theta}_{\mathrm{rg}} - \theta^\star_{\mathrm{rg}}\|_2 \leq \|\widehat{\theta}_{\mathrm{rg}} - \tilde{\theta}_{\mathrm{rg}}\|_2 + \|\tilde{\theta}_{\mathrm{rg}} - \theta^\star_{\mathrm{rg}}\|_2 \leq 2\sqrt{\frac{\epsilon_1}{L_1}},$$

hence

$$\|\widehat{\theta}_{\mathrm{rg}} - \theta^\star_{\mathrm{rg}}\|_2^2 \leq \frac{4\,\epsilon_1}{L_1}.$$

Finally,

$$\|\widehat{\theta}_{\mathrm{rg}} - \theta^\star\|_2 \geq \|\theta^\star_{\mathrm{rg}} - \theta^\star\|_2 - \|\widehat{\theta}_{\mathrm{rg}} - \theta^\star_{\mathrm{rg}}\|_2 \geq \delta - \sqrt{\frac{4\,\epsilon_1}{L_1}},$$

and if $\epsilon_1 \leq L_1\delta^2/16$, then $\sqrt{4\epsilon_1/L_1} \leq \delta/2$, which gives

$$\|\widehat{\theta}_{\mathrm{rg}} - \theta^\star\|_2 \geq \frac{\delta}{2}.$$

$\square$

*Remark* B.3. Notice that when we take $\epsilon_1 \leq L_1\delta^2/16$, we will have

$$\|\widehat{\theta}_{J,K} - \theta^\star_J\|_2^2 \leq 4\epsilon_1/L_1 < \frac{\delta}{2} \leq \|\widehat{\theta}_{\mathrm{rg}} - \theta^\star\|_2.$$

This directly yields Theorem 4.1.

## C    EVALUATED MODELS

We evaluate a series of state-of-the-art LVLMs and Multi-agent. The single-agent models include LLaVA (Liu et al., 2024a), Yi-VL-34B (Young et al., 2024), Qwen-VL (Bai et al., 2025), LLaVA-Med (Li et al., 2023), MedFlamingo (Moor et al., 2023), RadFM (Wu et al., 2023), HuatuoGPT-Vision (Chen et al., 2024a) and GPT-4o (OpenAI, 2024). The multi-agent frameworks include prior collaborative systems such as MedAgents (Tang et al., 2024), MDAgents (Kim et al., 2024) and AFlow (Zhang et al., 2024), as well as our proposed MMedAgent-RL framework that introduces reinforcement learning for adaptive multi-agent reasoning.

- **GPT-4o** (OpenAI, 2024) is OpenAI's latest multimodal large model that supports text, image, and audio inputs. It exhibits strong generalization across vision-language benchmarks and serves both as a single-agent baseline and as a specialist in our multi-agent settings.

- **Med-Flamingo** (Moor et al., 2023) is a multimodal few-shot learner designed for the medical domain. Built upon OpenFlamingo, it is further pre-trained on biomedical image-text data from scientific literature. It enables few-shot medical visual question answering with minimal supervision.

- **RadFM** (Wu et al., 2023) is a domain-specific foundation model tailored for radiology. It leverages large-scale radiology reports and domain-adaptive learning to improve zero-shot and few-shot performance on radiographic image understanding.

- **LLaVA-Med** (Li et al., 2023) extends LLaVA to the biomedical domain by fine-tuning with medical image-instruction pairs. It enhances medical reasoning and answer generation with limited supervision using domain-specific visual-textual alignments.

- **Qwen2.5-VL** (Bai et al., 2025) is a versatile vision-language model developed by Alibaba. It supports high-quality OCR, multi-turn dialogue, and reasoning over complex multimodal inputs. It is used both as a strong single-agent baseline and as the foundation of agents in our proposed framework.

- **Yi-VL-34B** (Young et al., 2024) is a large-scale multimodal model from 01.AI. With 34 billion parameters, it offers high-capacity visual understanding and serves as a powerful open-source baseline across medical and general VQA tasks.

- **LLaVA** (Liu et al., 2024b;a) are general-purpose vision-language models trained via visual instruction tuning. Evaluated in several sizes (7B, 13B, 34B), they serve as strong single-agent baselines in both in-domain and out-of-domain medical benchmarks.

- **HuatuoGPT-Vision-7B** (Chen et al., 2024a) is a medical multimodal large language model (MLLM) trained on the curated PubMedVision dataset. This dataset was created by using GPT-4V to denoise and reformat 1.3 million image-text pairs from PubMed, significantly improving data quality. As a result, HuatuoGPT-Vision demonstrates superior performance on medical multimodal benchmarks compared to other open-source models.

## D    EVALUATED DATASETS

We employ three established medical vision-language datasets: VQA-RAD (Lau et al., 2018), SLAKE (Liu et al., 2021), and PathVQA (He et al., 2020). Furthermore, to evaluate out-of-distribution performance, we incorporate the health and medicine subset of MMMU (Yue et al., 2024) along with OmniMedVQA (Hu et al., 2024b).

- **VQA-RAD** (Lau et al., 2018) is a manually constructed dataset containing 315 radiology images with 3,515 question-answer pairs. The images are distributed across head, chest, and abdomen regions, and include both open-ended and binary "yes/no" questions. Each image is associated with multiple clinically relevant questions generated by medical professionals. The dataset aims to facilitate the development of visual question answering systems for the medical domain.

- **SLAKE** (Liu et al., 2021) is a semantically-labeled knowledge-enhanced dataset featuring 642 radiology images and over 14,000 question-answer pairs. It offers comprehensive annotations including masks for semantic segmentation and bounding boxes for object detection. SLAKE is bilingual (English and Chinese) and covers 12 diseases and 39 organs across various body parts. The dataset also incorporates a medical knowledge graph with 5,232 medical knowledge triplets to support knowledge-based reasoning.

- **PathVQA** (He et al., 2020) is a pathology-focused dataset containing 32,799 open-ended questions from 4,998 pathology images. The dataset was created using a semi-automated pipeline to extract images and captions from pathology textbooks and generate question-answer pairs using natural language processing. PathVQA aims to support the development of AI systems capable of answering clinical questions about pathology images, with each question manually checked for correctness.

- **MMMU** (Yue et al., 2024) (Health & Medicine subset) is part of the Massive Multi-discipline Multimodal Understanding benchmark. This subset contains approximately 1,752 test questions across five disciplines: Basic Medical Science, Clinical Medicine, Diagnostics and Laboratory Medicine, Pharmacy, and Public Health. The questions require college-level subject knowledge and deliberate reasoning, challenging models to perform expert-level perception and reasoning tasks.

- **OmniMedVQA** (Hu et al., 2024b) is a comprehensive medical VQA benchmark collected from 73 different medical datasets, featuring images across 12 different modalities and covering more than 20 distinct anatomical regions. All images are sourced from authentic medical scenarios, ensuring alignment with real-world applications. The benchmark provides a diverse evaluation platform for testing the capabilities of large vision-language models in medical image understanding and reasoning.

## E  OVERVIEW OF THE BASELINES

We evaluate MMedAgent-RL against two main multi-agent baselines, MedAgents (Tang et al., 2024), MDAgents (Kim et al., 2024) and AFlow (Zhang et al., 2024). These baselines represent state-of-the-art approaches in medical visual question answering.

- **MedAgents** (Tang et al., 2024) establishes a zero-shot multi-agent collaboration framework that simulates real-world clinical workflows. The framework encompasses five critical steps: gathering domain experts, proposing individual analyses, summarizing analyses into a report, iterating over discussions until consensus is reached, and making a final decision. Different agents are assigned specific medical roles and collaborate to solve complex medical reasoning tasks. The framework relies on pre-trained large language models without additional fine-tuning, enabling natural dialogue-based interactions between agents. MedAgents demonstrates how specialized medical knowledge from different domains can be integrated through structured agent collaboration, providing a strong baseline for multi-agent medical reasoning.

- **MDAgents** (Kim et al., 2024) advances multi-agent medical systems by introducing adaptive collaboration mechanisms. Unlike fixed collaboration patterns, MDAgents dynamically selects the most appropriate agent configuration and communication structure based on the specific medical task. This framework allows for more flexible interactions between general practitioners and specialist agents, optimizing the collaboration pattern for different types of medical queries. MDAgents incorporates mechanisms to resolve conflicts between different agent opinions and adapts the consultation workflow to match the complexity of the medical case, resulting in more robust decision-making across diverse medical scenarios.

- **AFlow** (Zhang et al., 2024) is a framework designed to automatically generate and optimize complex problem-solving workflows for LLMs. Instead of relying on a single inference pass, these workflows enhance performance through structured procedures. We evaluate three distinct strategies as strong baselines: *Self-Consistency Ensemble*, which runs an agent multiple times and selects the most frequent answer to improve reliability; *Multi-Agent Debate*, which uses several agents to collaboratively propose, critique, and refine solutions; and *Self-Refine*, which employs a feedback loop for a single agent to iteratively critique and improve its own output.

## F  EXPERIMENTAL SETUP

### F.1  DATA STATISTICS

The data used in this work is shown in Table 3 and involves five multimodal medical datasets: VQA-RAD, SLAKE, PathVQA, OmniMedVQA and MMMU (Health & Medicine track). Among them, three are used as in-domain datasets, with their training sets employed for model training. The

Table 3: The results of the medical VQA benchmark. Here, MMMU denotes MMMU (Health & Medicine track) and the number of training and testing phase denotes the number of QA items for each phase.

| Model | All | VQA-RAD | SLAKE | PathVQA | OmniMedVQA | MMMU |
|---|---|---|---|---|---|---|
| Train | 12,176 | 940 | 1,681 | 9,555 | / | / |
| - Easy | 8,321 | 498 | 1,284 | 6,539 | / | / |
| - Medium | 1,409 | 160 | 114 | 1,135 | / | / |
| - Hard | 2,626 | 281 | 275 | 2,070 | / | / |
| Test | 15,153 | 251 | 416 | 3,362 | 11,124 | 150 |

remaining two are directly used as out-of-domain (OOD) testing datasets. The specific data volume for each dataset used at each stage of Curriculum-Based Multi-Agent Reinforcement Learning is detailed in Table 3.

## F.2 HYPERPARAMETER SETTINGS

We use Qwen2.5-VL (Bai et al., 2025) as the base model. We design the prompt template using the format employed in MM-EUREKA (Meng et al., 2025), clearly specifying the required output structure, which includes using `<think>` and `<answer>` tags to separately contain the reasoning process and the final answer, with the two being separated. The detailed prompt is shown in Table 4. For training hyperparameters, the rollout batch size and training batch size are both set to 128, with 8 rollouts generated for each sample. The sampling temperature is set to 1.0 to encourage response diversity, and optimization is done with a learning rate of $1 \times 10^{-6}$. Additionally, for the three stages of curriculum reinforcement learning, the KL divergence coefficients are set to $1 \times 10^{-3}$, $4 \times 10^{-3}$, and $1 \times 10^{-2}$ respectively to stabilize training. The dynamic entropy coefficient $\gamma_s$ is set to 0.03 for hard ($s = 0$), 0.005 for medium ($0 < s < 1$), and 0.0001 for easy ($s = 1$) samples, respectively, to adapt the level of exploration based on curriculum difficulty. For the number of specialists, we set $e = 3$. For the baseline implementation, i.e., MedAgents (Tang et al., 2024), MDAgents (Kim et al., 2024) and AFlow (Zhang et al., 2024), we use Qwen2.5-VL as the agent for decision making to ensure a fair comparison between multi-agent baselines and MMedAgent-RL. For the training framework, we adopt a multimodal RL framework based on OpenRLHF (Hu et al., 2024a). For the inference, we adopt the vLLM framework (Kwon et al., 2023). All training is conducted on 8 NVIDIA Tesla A100 80GB GPUs.

## F.3 PROMPT

The prompt for the fine-tuning of base model is shown in Table 4. In this prompt, we provide the question options, the input image, and $k$ expert answers. In the experiment, $k$ is set to 3. The model needs to first generate the reasoning process within the `<think>` tag, and then provide the final answer within the `<answer>` tag.

Table 4: Prompt template used for reinforcement learning fine-tuning.

> **Prompt Template:**
> As the General Practitioner coordinating this case, review the specialist expertise to make a final decision. Answer from `<Specialist>`: `<SpecialistAnswer>`. `<Question>` Provide your final assessment. You need to first think about the reasoning process in the mind and then provide the user with the answer. The reasoning process and answer are enclosed within `<think> </think>` and `<answer> </answer>` tags, respectively, i.e., `<think>` reasoning process here `</think><answer>` answer here `</answer>`. The answer must be chosen from the given options.

Table 5: The comparison with SFT on several medical VQA benchmarks.

| Model | In-Domain Datasets | | | | Out-of-Distribution Datasets | | |
|---|---|---|---|---|---|---|---|
| | VQA-RAD | SLAKE | PathVQA | Avg. | OmniMedVQA | MMMU-Med | Avg. |
| GPT-4o | 61.0 | 75.5 | 69.4 | 68.6 | 68.5 | 69.7 | 69.1 |
| Med-Flamingo | 45.4 | 43.5 | 54.7 | 47.9 | 30.7 | 28.3 | 29.5 |
| RadFM | 50.6 | 34.6 | 38.7 | 41.3 | 28.2 | 27.0 | 27.6 |
| LLaVA-Med-7B | 51.4 | 48.6 | 56.8 | 52.3 | 44.1 | 36.9 | 40.5 |
| Qwen-VL-Chat | 47.0 | 56.0 | 55.1 | 52.7 | 48.3 | 32.7 | 40.5 |
| Yi-VL-34B | 53.0 | 58.9 | 47.3 | 53.1 | 51.5 | 41.5 | 46.5 |
| LLaVA-v1.6-7B | 52.6 | 57.9 | 47.9 | 52.8 | 49.0 | 33.1 | 41.1 |
| LLaVA-v1.6-13B | 55.8 | 58.9 | 51.9 | 55.5 | 48.0 | 39.3 | 43.7 |
| LLaVA-v1.6-34B | 58.6 | 67.3 | 59.1 | 61.6 | 58.7 | 48.8 | 53.8 |
| LLaVA-v1.5-LLaMA3-8B | 54.2 | 59.4 | 54.1 | 55.9 | 44.6 | 38.2 | 41.4 |
| HuatuoGPT-Vision-7B | 63.0 | 77.2 | 58.7 | 66.3 | 74.6 | 51.0 | 62.8 |
| Qwen2.5-VL-3B | 61.0 | 62.7 | 57.6 | 60.4 | 60.1 | 54.5 | 57.3 |
| Qwen2.5-VL-7B | 61.8 | 64.7 | 60.5 | 62.3 | 60.8 | 56.6 | 58.7 |
| MedVLThinker-7B | 63.7 | 67.8 | 65.2 | 65.6 | 62.4 | 57.0 | 59.7 |
| **Multi-Agent Collaboration** | | | | | | | |
| MedAgents | 65.6 | 67.9 | 63.2 | 65.6 | 55.8 | 49.7 | 52.6 |
| MDAgents | 66.8 | 68.2 | 65.4 | 66.8 | 58.2 | 52.3 | 55.1 |
| AFlow | 67.3 | 68.9 | 66.4 | 67.5 | 59.6 | 53.6 | 56.6 |
| GPT-4o → Qwen2.5-VL-7B | 62.5 | 63.9 | 53.2 | 59.9 | 56.4 | 50.7 | 53.6 |
| GPT-4o → Qwen2.5-VL-7B+SFT w/o reasoning | 65.5 | 66.5 | 61.4 | 64.5 | 60.9 | 57.8 | 62.4 |
| GPT-4o → Qwen2.5-VL-7B+SFT w/ reasoning | 68.8 | 68.4 | 63.7 | 67.0 | 62.0 | 59.4 | 64.5 |
| **MMedAgent-RL** (7B) | 71.5 +10% | 76.2 +12% | 72.3 +12% | 73.3 +11% | 73.3 +13% | 71.9 +15% | 72.6 +14% |

## F.4 DATASET SPLITTING AND DIFFICULTY STRATIFICATION

The partition of training, validation, and test sets strictly follows the official dataset guidelines. The difficulty grading is required exclusively for the training phase to implement curriculum learning. Specifically, we classify samples into difficulty levels based on the consistency and accuracy of the specialists' responses. We emphasize that the inference process and the calculation of evaluation scores do not use this difficulty grading at all. The model generates responses without access to any difficulty labels or ground truth. The difficulty stratification on the test set is applied solely for the analytical breakdown shown in Figure 4. It serves only to visualize and analyze model performance across different complexity levels post-generation, and has no influence on the inference process itself. The difficulty grading for the test dataset follows the same as training data.

## G ADDITIONAL RESULTS

### G.1 COMPARISON WITH SUPERVISED FINE-TUNING (SFT)

We compare our method with Supervised Fine-Tuning (SFT) methods as a baseline. As shown in Table 5, our proposed method, MMedAgent-RL, demonstrates a significant performance advantage over SFT methods across all the medical VQA benchmarks. The superiority of our approach is even more pronounced in the more challenging out-of-distribution datasets. This highlights our model's enhanced robustness and generalization capabilities. Overall, MMedAgent-RL consistently sets a new state-of-the-art, with our base model achieving a 73.3% average on in-domain tasks and 72.6% on out-of-distribution tasks, already surpassing the SFT method. The results clearly indicate that our multi-agent, reinforcement learning-based approach is more effective than traditional SFT techniques for complex medical VQA tasks.

### G.2 DIFFERENT SPECIALISTS

The effectiveness of different specialist compositions within our framework is detailed in Table 6. The results unequivocally show that multi-agent collaboration substantially outperforms single-agent baselines. Our premier configuration, `3*OpenAI-o3` (OpenAI, 2025), achieved a top average score of 73.0, far exceeding the best-performing baseline, GPT-4o (68.8). Crucially, this performance gain stems from the synergistic integration of multiple experts. The analysis also underscores the importance of specialist diversity. Heterogeneous teams combining different models (e.g., `2*OpenAI-o3+1*HuatuoGPT` at 71.3) proved highly effective, demonstrating that fusing complementary knowledge enhances diagnostic robustness. This confirms that our framework's core strength lies in its ability to dynamically orchestrate collaboration among a diverse team of high-quality specialists to achieve superior decision-making.

Table 6: The comparison with different specialists.

| Model | VQA-RAD | SLAKE | PathVQA | OmniMedVQA | MMMU | Avg. |
|---|---|---|---|---|---|---|
| GPT-4o | 61.0 | 75.5 | 69.4 | 68.5 | 69.7 | 68.8 |
| Qwen2.5-VL-7B | 61.8 | 64.7 | 60.5 | 60.8 | 56.6 | 60.9 |
| HuatuoGPT-7B | 63.0 | 77.2 | 58.7 | 74.6 | 51.0 | 64.9 |
| **MMedAgent-RL** | | | | | | |
| 3*OpenAI-o3 | 71.5 | 76.2 | 72.3 | 73.3 | 71.9 | 73.04 |
| 3*GPT-4o | 70.4 | 75.2 | 72.7 | 69.1 | 67.1 | 70.9 |
| 1*GPT-4o | 63.2 | 65.9 | 62.8 | 63.4 | 59.0 | 62.86 |
| 1*HuatuoGPT-Vision-7B | 63.4 | 67.2 | 60.8 | 64.9 | 54.1 | 62.1 |
| 1*Qwen2.5-VL-7B | 62.4 | 63.2 | 61.5 | 61.5 | 55.1 | 60.7 |
| 2*GPT-4o+1*Qwen2.5-VL-7B | 70.0 | 74.6 | 71.3 | 66.0 | 64.0 | 69.2 |
| 2*GPT-4o+1*HuatuoGPT-Vision-7B | 68.5 | 75.0 | 71.7 | 68.0 | 62.0 | 69.0 |
| 2*OpenAI-o3+1*HuatuoGPT-Vision-7B | 69.9 | 75.8 | 73.0 | 70.0 | 68.0 | 71.3 |
| 2*OpenAI-o3+1*Qwen2.5-VL-7B | 71.1 | 75.4 | 71.8 | 69.1 | 69.2 | 71.3 |
| 1*Qwen2.5-VL-7B+1*GPT-4o | 68.9 | 73.8 | 70.6 | 66.1 | 63.2 | 68.5 |
| 1*HuatuoGPT-Vision-7B+1*OpenAI-o3 | 69.0 | 74.7 | 72.3 | 68.8 | 68.3 | 70.6 |
| 3*HuatuoGPT-Vision-7B | 65.8 | 78.2 | 61.3 | 73.8 | 50.1 | 65.8 |

## G.3 ABLATION ANALYSIS

### G.3.1 PERFORMANCE OF TRIAGE DOCTOR

The accuracy of the triage doctors is shown in Table 7. We used the data with definitive department labels as the evaluation target. From the results, we can observe that triage is not as challenging as answering complex medical diagnostic questions. Instead, department classification resembles a modality classification process. The original model already achieved an accuracy of over 80%, and after our fine-tuning, the model's performance has reached a human-level standard on these datasets.

Table 7: The performance of triage doctor.

| Model | VQA-RAD | SLAKE | PathVQA |
|---|---|---|---|
| Qwen2.5-VL-3B | 95.62 | 92.16 | 77.53 |
| Qwen2.5-VL-7B | 96.21 | 94.41 | 80.58 |
| **MMedAgent-RL** | 99.98 | 99.94 | 99.06 |

### G.3.2 KL DIVERGENCE COEFFICIENT

We conduct ablation experiments on the KL divergence coefficient at each stage, and the results are shown in Figure 6. We observe that in the first stage, as the KL divergence coefficient increases, the model's performance tends to stabilize. This indicates that when training with simple data, where the specialist doctor's answers are entirely correct, i.e., the model merely needs to learn to imitate. In this case, an additional KL divergence loss is required to constrain the policy model's update steps, preventing it from changing too drastically; otherwise, it would become a model that simply copies the specialist's answers. In the second stage, the optimal KL divergence coefficient is slightly larger than in the first stage, suggesting that the model needs some autonomy to explore its own direction. This becomes even more apparent in the third stage, where the optimal KL divergence coefficient is significantly higher. This is reasonable because, when the specialist doctor's answers are entirely incorrect, it becomes very difficult for the model to generate an accurate response. If the KL divergence loss is too large in this stage, the model cannot explore effectively to find the correct answer. Therefore, in conclusion, different KL divergence coefficients need to be set for each stage of curriculum reinforcement learning to ensure optimal model performance.

## G.4 COMPARISON WITH TEST-TIME SCALING METHODS

We conducted additional experiments using test-time scaling techniques, specifically Majority Voting and Self-Consistency (Wang et al.), on both Qwen2.5-VL-7B and GPT-4o. For the experimental setup, Majority Voting involved sampling $N = 3$ outputs and selecting the most frequent answer as the final prediction. Self-Consistency sampled a set of diverse reasoning paths rather than relying

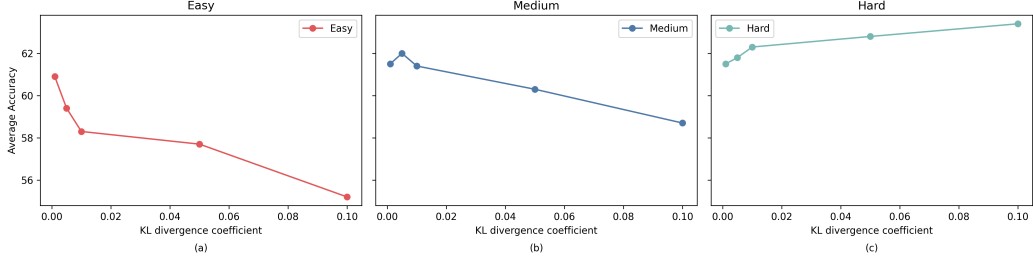

Figure 6: Ablation of KL divergence coefficient.

on the greedy path, subsequently selecting the most consistent answer by marginalizing over the sampled paths. As shown in Table 8, test-time scaling improved the performance of Qwen2.5-VL-7B by 2.6% and GPT-4o by 1.6% across the five datasets. However, despite these improvements, a performance gap remains compared to our multi-agent framework. Our method outperforms these enhanced baselines by 18%, demonstrating the necessity and effectiveness of our proposed triage and multi-expert pipeline.

Table 8: Performance comparison with test-time scaling baselines.

| Method | VQA-RAD | SLAKE | PathVQA | OmniMedVQA | MMMU-Med | Overall |
|---|---|---|---|---|---|---|
| Qwen2.5-VL-7B | 61.8 | 64.7 | 60.5 | 60.8 | 56.6 | 60.9 |
| Qwen2.5-VL-7B + Majority Voting | 63.7 | 65.4 | 61.5 | 62.5 | 57.0 | 62.0 |
| Qwen2.5-VL-7B + Self-consistency | 63.8 | 65.3 | 62.4 | 64.5 | 59.1 | 63.0 |
| GPT-4o | 61.0 | 75.5 | 69.4 | 68.5 | 69.7 | 68.8 |
| GPT-4o + Majority Voting | 62.2 | 75.9 | 70.3 | 69.4 | 70.3 | 69.6 |
| GPT-4o + Self-consistency | 63.6 | 75.4 | 70.8 | 69.1 | 71.5 | 70.1 |
| **MMedAgent-RL** | **71.5** | **76.2** | **72.3** | **73.3** | **71.9** | **73.0** |

### G.5 QUANTIFY THE PERFORMANCE GAIN FROM ROUTING AND AGGREGATION

To strictly quantify the model's ability to correct mistakes rather than simply route to a specialist, we conducted a targeted evaluation on hard samples where all three specialists provided incorrect answers. We selected 200 such samples from PathVQA and 200 from OmniMedVQA to evaluate the performance of the Base Model (with and without expert knowledge), the SFT baseline, and our MMedAgent-RL. Since every specialist is incorrect, any routing mechanism would yield 0% accuracy. Therefore, the performance on this subset is strictly attributable to the model's ability to perform intrinsic correction. As shown in Table 9, our method achieves substantial improvements. For instance, on the OOD OmniMedVQA hard subset, MMedAgent-RL improves accuracy to 23.0%. We attribute this capability to the following logic: The extremely low accuracy of the Base Model w/o expert knowledge (4.5% on PathVQA and 2.0% on OmniMedVQA) confirms that the model lacks the intrinsic parametric knowledge to solve these hard cases independently. Although the specialists' final answers were wrong, their reasoning processes likely contained partial truths or valid medical context. While the SFT baseline struggles to utilize this conflicting information (often hallucinating along with the experts), MMedAgent-RL has learned via RL to critically synthesize these valid reasoning fragments, correcting the final conclusion rather than simply aggregating the errors.

### G.6 DETAILED ABLATION ON TRIAGE AGENT

In Table 2, "w/o Triage" refers to using the original base model to perform the routing (triage) task directly, without specific fine-tuning for this role. Following your suggestions, we have expanded our comparison in Table 10. We introduced two new settings to test the necessity of the framework: 1) Random Routing: Assigning queries to specialists randomly to isolate the benefit of the specialists themselves. 2) Single Model w/o Routing: Using the Qwen2.5-VL-7B base model directly, and enhancing it with Majority Voting (diverse sampling) as requested.

As shown in Table 10, incorporating a routing mechanism leads to significant performance gains. No-

Table 9: Correctness ratio (accuracy on hard samples where all specialists failed).

| Setting / Model | PathVQA | OmniMedVQA |
|---|---|---|
| *w/o expert knowledge* | | |
| Base Model | 4.5% | 2.0% |
| *w/ expert knowledge* | | |
| Base Model | 5.5% | 3.5% |
| SFT | 10.0% | 7.5% |
| **MMedAgent-RL** | **26.5%** | **23.0%** |

tably, even Random Routing generally outperforms the single model equipped with Majority Voting. Furthermore, our proposed MMedAgent-RL significantly outperforms all baselines, confirming that a dedicated triage-and-expert pipeline provides advantages that cannot be achieved by simple diverse sampling or random assignment.

**Comparison with SFT.** The original "w/o Triage" ablation was insufficient to justify the specific choice of RL over simpler methods. To address this, we conducted a detailed comparison between our GRPO-optimized Triage Agent and standard Supervised Fine-Tuning (SFT) baselines. As shown in Table 11, we evaluated two SFT configurations: 1) SFT (Standard): Fine-tuned on direct Question-Department pairs. 2) SFT (w/ Reasoning): Fine-tuned using reasoning traces distilled from Qwen2.5-VL-32B to simulate a more capable classifier. While SFT significantly improves performance over the base model, GRPO still outperforms the best SFT baseline across all datasets. Crucially, this advantage is most pronounced on Out-of-Distribution (OOD) datasets. For instance, on MMMU-Med, GRPO outperforms "SFT w/ Reasoning" by +4.1% (71.9 vs. 67.8), and on OmniMedVQA by +2.5% (73.3 vs. 70.8). This indicates that while SFT can achieve high accuracy on standard distributions, the reinforcement learning process (GRPO) enables the Triage Agent to generalize better to complex, unseen scenarios by learning from the reward signal of the downstream reasoning success, rather than just mimicking a static label.

Table 10: Performance comparison with or without routing.

| Method | VQA-RAD | SLAKE | PathVQA | OmniMedVQA | MMMU-Med | Overall |
|---|---|---|---|---|---|---|
| *w/ triage* | | | | | | |
| MMedAgent-RL (zero-shot) | 66.3 | 69.9 | 67.2 | 66.2 | 59.3 | 65.8 |
| MMedAgent (random triage) | 63.4 | 66.7 | 64.7 | 67.8 | 58.2 | 64.2 |
| **MMedAgent-RL (fine-tuned triage)** | **71.5** | **76.2** | **72.3** | **73.3** | **71.9** | **73.0** |
| *w/o triage* | | | | | | |
| Qwen2.5-VL-7B | 61.8 | 64.7 | 60.5 | 60.8 | 56.6 | 60.9 |
| + Majority Voting | 63.7 | 65.4 | 61.5 | 62.5 | 57.0 | 62.0 |

Table 11: Performance comparison of Triage Agent training strategies.

| Method | VQA-RAD | SLAKE | PathVQA | OmniMedVQA | MMMU-Med |
|---|---|---|---|---|---|
| Base model | 66.3 | 69.9 | 67.2 | 66.2 | 59.3 |
| + SFT w/o reasoning process | 69.4 | 75.0 | 70.5 | 70.1 | 66.7 |
| + SFT w/ reasoning process | 70.2 | 75.9 | 71.0 | 70.8 | 67.8 |
| + GRPO | **71.5** | **76.2** | **72.3** | **73.3** | **71.9** |

## G.7 DETAILED ABLATION ON PROGRESSIVELY ADDING COMPONENTS

To quantify the contribution of each component, we have conducted a progressive evaluation as shown in Table 12. Regarding the order of ablation, we formulated the progression as Base → Multi-expert → Triage. From an architectural perspective, the Triage module depends on the existence of a candidate pool of experts to perform routing. Thus, we first introduce the Specialists (Multi-expert) to build the capability pool, and subsequently add the Triage module to manage and utilize these experts efficiently. Starting with the Qwen2.5-VL-7B baseline, we observed the following trends:

- + Majority Voting: Provides a marginal improvement, indicating that simple test-time scaling has limits.
- + Specialists: Integrating domain-specific experts (with a base model as the router) yields further gains, surpassing the single model with voting.
- + Triage: Introducing the learned Triage module significantly improves the effective utilization of specialists.
- + Curriculum RL: Finally, applying our Curriculum RL strategy provides the most substantial performance leap, demonstrating that optimizing the collaboration between the triage and specialist agents is critical for complex medical reasoning.

Table 12: Performance progressively adding components.

| Method | VQA-RAD | SLAKE | PathVQA | OmniMedVQA | MMMU-Med |
|---|---|---|---|---|---|
| Qwen2.5-VL-7B (Base) | 61.8 | 64.7 | 60.5 | 60.8 | 56.6 |
| + Specialists (Base Model) | 64.5 | 66.9 | 63.2 | 63.4 | 60.7 |
| + Triage | 65.7 | 68.4 | 64.4 | 64.8 | 62.6 |
| + Curriculum RL | **71.5** | **76.2** | **72.3** | **73.3** | **71.9** |

## G.8 Comparison on Triage Agent with Different Settings

As shown in Table 13, we quantitatively evaluated the impact of different training stages on the triage agent's performance. We compared the Base Model, SFT (trained on direct question-answer pairs without reasoning), SFT with Reasoning (trained on reasoning traces distilled from Qwen2.5-VL-32B), and our final GRPO-optimized model. The results demonstrate that the triage agent trained with GRPO yields the highest performance. Notably, this improvement is most significant on the two Out-of-Distribution (OOD) datasets, i.e., OmniMedVQA and MMMU-Med, confirming that the reasoning capabilities reinforced by GRPO are crucial for generalization.

Table 13: Performance with triage agent with different settings.

| Method | VQA-RAD | SLAKE | PathVQA | OmniMedVQA | MMMU-Med | Overall |
|---|---|---|---|---|---|---|
| Base model | 66.3 | 69.9 | 67.2 | 66.2 | 59.3 | 65.8 |
| + SFT w/o reasoning process | 69.4 | 75.0 | 70.5 | 70.1 | 66.7 | 70.3 |
| + SFT w/ reasoning process | 70.2 | 75.9 | 71.0 | 70.8 | 67.8 | 71.1 |
| + GRPO | **71.5** | **76.2** | **72.3** | **73.3** | **71.9** | **73.0** |

## G.9 Comparison on GP Update Strategies

In the paper, the two GP agents (the Triage Agent and the Attending Physician) are updated independently. This design choice was primarily made to ensure training stability and to decouple the training process from potential API failures or latency when querying the external OpenAI-based specialists. Specifically, our procedure is as follows: 1) Triage Optimization: We first optimize the Triage Agent using image-modality QA pairs to ensure accurate department routing. 2) Data Preparation: We classify the training data based on these departments and invoke the OpenAI API (acting as specialists) to generate expert knowledge offline. 3) Attending GP Training: Finally, we use these pre-generated expert trajectories to train the Attending GP (Qwen2.5-VL) via Reinforcement Learning.

We also implemented an end-to-end online framework where both GPs are updated simultaneously. As shown in Table 14, the performance difference between the two settings is negligible. This confirms that our decoupled training strategy is valid and yields results consistent with a fully end-to-end approach while remaining more computationally efficient and stable.

## G.10 Framework Transferability

To demonstrate the transferability of our framework, we conducted additional experiments using InternVL2.5-Instruct-8B (Chen et al., 2024b) as an alternative base model. As shown in Table 15, our

Table 14: Performance comparison of GP update strategies.

| Strategy | VQA-RAD | SLAKE | PathVQA | OmniMedVQA | MMMU-Med | Overall |
|----------|---------|-------|---------|------------|----------|---------|
| Independent | **71.5** | 76.2 | 72.3 | 73.3 | **71.9** | 73.0 |
| Simultaneous | 71.3 | **76.5** | **72.4** | **73.6** | 71.6 | **73.1** |

method yields consistent and significant improvements across all datasets, regardless of the backbone architecture. On OmniMedVQA, the InternVL-based agent achieved a remarkable score of 82.4%, surpassing the performance of the Qwen-based version. Even on datasets where the base InternVL model struggled (e.g., PathVQA, where the base score was only 42.3%), our framework provided a massive performance boost of +26.1% (reaching 68.4%). These results confirm that our pipeline is model-agnostic and can effectively enhance the reasoning capabilities of diverse multimodal LLMs.

Table 15: Assessment of framework transferability.

| Model | VQA-RAD | SLAKE | PathVQA | OmniMedVQA | MMMU-Med |
|-------|---------|-------|---------|------------|----------|
| Qwen2.5-VL-7B | 61.8 | 64.7 | 60.5 | 60.8 | 56.6 |
| **MMedAgent-RL (Qwen2.5-VL-7B)** | **71.5** | **76.2** | **72.3** | **73.3** | **71.9** |
| InternVL2.5-8B | 58.6 | 68.6 | 42.3 | 76.5 | 51.4 |
| **MMedAgent-RL (InternVL2.5-8B)** | **70.2** | **78.9** | **68.4** | **82.4** | **64.7** |

## G.11 BASELINES WITH TRAINED AGGREGATOR

We implemented two new baselines where GPT-4o first samples $N = 3$ diverse outputs per query ($T = 1.0$), and a Qwen2.5-VL-7B model is then trained on the training set (same as data we used) to act as an aggregator that selects or synthesizes the final answer from these candidates. We developed both an SFT Aggregator (via Supervised Fine-Tuning) and a GRPO Aggregator (via Group Relative Policy Optimization) to ensure a robust comparison. As shown in Table 16, MMedAgent-RL still maintains a significant performance lead. We observed that directly training an aggregator yields limited gains, particularly when facing inconsistent candidate answers, i.e., a challenge that directly motivated our proposed curriculum learning-guided RL strategy. This confirms that our method's effectiveness stems from the process-level collaboration of specialized agents, which captures domain-specific nuances that cannot be replicated by simply aggregating generalist outputs.

Table 16: Performance comparison with GPT-4o+an aggregator (based on Qwen2.5-VL 7B) baselines.

| Model | VQA-RAD | SLAKE | PathVQA | OmniMedVQA | MMMU-Med | Overall |
|-------|---------|-------|---------|------------|----------|---------|
| Qwen2.5-VL-7B (Base) | 61.8 | 64.7 | 60.5 | 60.8 | 56.6 | 60.9 |
| GPT-4o | 61.0 | 75.5 | 69.4 | 68.5 | 69.7 | 68.8 |
| GPT-4o+Qwen2.5-VL-7B (SFT Fine-tuned) | 65.8 | 69.4 | 62.9 | 60.3 | 65.3 | 64.7 |
| GPT-4o+Qwen2.5-VL-7B (GRPO Fine-tuned) | 67.3 | 70.8 | 63.8 | 61.8 | 64.6 | 65.7 |
| **MMedAgent-RL** | **71.5** | **76.2** | **72.3** | **73.3** | **71.9** | **73.0** |

## G.12 DETAILED RESULTS

**Traditional Medical Imaging Evaluation.** Table 17 presents the accuracy of various models across five major medical imaging modalities in the OmniMedVQA benchmark. Our model (MMedAgent-RL) demonstrates strong generalization across all categories, achieving an average accuracy of 73.3%, significantly outperforming previous state-of-the-art models including LLaVA-v1.6-34B (58.7%) and Qwen2.5-VL-7B (60.8%). Specifically, our method achieves 76% on microscopy images, indicating robust capability in processing fine-grained, high-resolution visual data typical of pathology slides. On MRI and CT modalities, MMedAgent-RL reaches 72% and 65%, respectively, outperforming strong baselines such as LLaVA-v1.6-34B and Yi-VL-34B by a wide margin. These results show that our model captures both structural and soft-tissue anatomical details effectively. In X-Ray, our method maintains competitive performance (78.8%) compared to high-performing models like HuatuoGPT-Vision-7B (80.3%), while achieving the highest accuracy on Ultrasound (75%) among

all models, demonstrating robustness in handling noisy, low-contrast imaging modalities.

**MMMU Health & Medicine Track.** In Table 18, our model again establishes new performance standards, achieving 71.9% overall accuracy on the MMMU Health & Medicine test set. Compared to existing large models such as Qwen2.5-VL-7B (56.6%) and HuatuoGPT-Vision-7B (51.0%), MMedAgent-RLdemonstrates clear advantages. Notably, our model excels across all five sub-domains: scoring 75% in Basic Medical Science (BMS), 78% in Clinical Medicine (CM), 65% in Diagnostics and Laboratory Medicine (DLM), 70% in Pharmacy (P), and 71.5% in Public Health (PH). These results reflect a well-rounded capability across both foundational scientific understanding and applied clinical knowledge. In particular, performance in CM and P shows substantial improvement over single-agent baselines, suggesting that our model benefits from enhanced reasoning and domain transfer. Taken together, these results confirm the effectiveness of our approach in both imaging-based and knowledge-based medical VQA settings, and highlight the potential of our method as a comprehensive solution for multimodal medical understanding.

Table 17: The accuracy of OmniMedVQA within different modalities (excluding FP, OCT, and Dermatology). **CT**: *Computed Tomography*, **MRI**: *Magnetic Resonance Imaging*, **Mic**: *Microscopy Images*, **X-Ray**: *X-ray*, **US**: *Ultrasound*.

| Model | CT | MRI | Mic | X-Ray | US | Avg. |
|---|---|---|---|---|---|---|
| Med-Flamingo | 34.6 | 27.5 | 28.1 | 30.1 | 33.2 | 30.7 |
| RadFM | 33.3 | 22.0 | 28.0 | 31.5 | 26.1 | 28.2 |
| LLaVA-Med-7B | 25.3 | 35.9 | 44.0 | 31.7 | 83.7 | 44.1 |
| Qwen-VL-Chat | 51.5 | 43.9 | 49.5 | 63.1 | 33.5 | 48.3 |
| Yi-VL-34B | 39.8 | 51.4 | 61.4 | 64.2 | 40.5 | 51.5 |
| LLaVA-v1.6-7B | 40.1 | 54.8 | 48.8 | 53.3 | 47.9 | 49.0 |
| LLaVA-v1.6-13B | 40.0 | 47.4 | 50.5 | 59.6 | 42.6 | 48.0 |
| LLaVA-v1.6-34B | 50.6 | 60.9 | 62.8 | 74.7 | 44.5 | 58.7 |
| LLaVA-v1.5-LLaMA3-8B | 33.0 | 53.8 | 48.4 | 56.6 | 31.2 | 44.6 |
| HuatuoGPT-Vision-7B | 65.6 | 72.7 | 77.5 | 80.3 | 76.7 | 74.6 |
| Qwen2.5-VL-3B | 60.5 | 64.2 | 66.6 | 68.9 | 40.4 | 60.1 |
| Qwen2.5-VL-7B | 62.0 | 68.3 | 70.7 | 68.9 | 34.3 | 60.8 |
| **Multi-Agent Collaboration** | | | | | | |
| MedAgents | 55.0 | 57.2 | 59.1 | 58.6 | 49.0 | 55.8 |
| MDAgents | 58.1 | 60.5 | 61.7 | 60.2 | 50.6 | 58.2 |
| AFlow | 59.5 | 62.0 | 63.2 | 61.7 | 51.6 | 59.6 |
| **MMedAgent-RL** (7B) | 64.6 | 71.7 | 76.0 | 78.8 | 75.4 | **73.3** |

## G.13 MORE CASES

To further demonstrate the robustness and versatility of our proposed model in multimodal medical applications, we present additional representative cases in Figure 7, Figure 8, Figure 9 and Figure 10. These examples encompass various clinical scenarios and imaging modalities, providing a comprehensive illustration of the model's ability to effectively integrate and interpret diverse types of medical data.

Table 18: Results on the test set for the MMMU Health & Medicine track. The Health & Medicine track is divided into five categories: **BMS** for *Basic Medical Science*, **CM** for *Clinical Medicine*, **DLM** for *Diagnostics and Laboratory Medicine*, **P** for *Pharmacy*, and **PH** for *Public Health*. Results are obtained by submitting to the official website.

| Model | BMS | CM | DLM | P | PH | MMMU Health & Medicine |
|---|---|---|---|---|---|---|
| Med-Flamingo | 29.6 | 28.1 | 24.8 | 25.3 | 31.2 | 28.3 |
| RadFM | 27.5 | 26.8 | 25.8 | 24.7 | 29.1 | 27.0 |
| LLaVA-Med-7B | 39.9 | 39.1 | 34.6 | 37.4 | 34.0 | 36.9 |
| Qwen-VL-Chat | 36.5 | 31.7 | 32.7 | 28.4 | 34.6 | 32.7 |
| Yi-VL-34B | 49.4 | 48.9 | 43.2 | 40.5 | 32.0 | 41.5 |
| LLaVA-v1.6-7B | 40.5 | 36.9 | 32.1 | 32.3 | 26.9 | 33.1 |
| LLaVA-v1.6-13B | 53.6 | 46.7 | 33.3 | 22.2 | 40.0 | 39.3 |
| LLaVA-v1.6-34B | 56.4 | 56.0 | 46.9 | 46.7 | 41.7 | 48.8 |
| HuatuoGPT-Vision-7B | 60.7 | 63.3 | 36.7 | 50.0 | 44.4 | 51.0 |
| Qwen2.5-VL-3B | 67.8 | 53.3 | 43.3 | 55.6 | 53.3 | 54.5 |
| Qwen2.5-VL-7B | 67.9 | 56.7 | 36.7 | 66.7 | 56.7 | 56.6 |
| **Multi-Agent Collaboration** | | | | | | |
| MedAgents | 60.0 | 55.0 | 45.0 | 50.0 | 52.9 | 52.6 |
| MDAgents | 62.5 | 57.5 | 50.0 | 55.0 | 50.5 | 55.1 |
| **MMedAgent-RL** (7B) | 75.0 | 78.0 | 65.0 | 70.0 | 71.5 | **71.9** |

---

**Question**: What can be observed in this image?
**Options**: A. Shin splints, B. Tennis elbow, C. Plantar fascia pathology, D. Meniscal tear
**Answer**: C. Plantar fascia pathology.

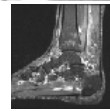

**Ours**: The image provided is a CT scan of a foot, specifically showing the plantar fascia. The CT scan typically shows bone structures and soft tissue, and in this case, it clearly displays the plantar fascia, which is a common site for plantar fasciitis or other plantar fascia-related pathologies.
<answer>C. Plantar fascia pathology</answer>

---

**Question**: The figure below <image 1> shows a stable structure of a protein that is to be used for each part of this problem. The dashed line divides the structure into three regions labeled A, B, and C in the figure. Does this protein have quaternary structure and how do you reach that conclusion?
**Options**: A. True, B. False.
**Answer**: B. False

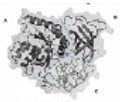

**Ours**: The figure shows a single protein structure with no indication of multiple protein subunits or different domains interacting. In proteins, quaternary structure refers to the arrangement of two or more polypeptide chains that form a complex. Since the image only displays one continuous chain without any visible subunit separation, it does not represent a quaternary structure.
<answer>B. False</answer>

Figure 7: Case 1 (CT scan of foot): The model identifies plantar fascia pathology from a CT image, requiring anatomical knowledge of soft tissue structures in the foot and recognition of characteristic changes consistent with plantar fasciitis. Case 2 (Protein structure diagram): A structural biology reasoning task based on a protein diagram, where the absence of distinct subunit boundaries leads to the conclusion that the protein does not exhibit quaternary structure—showcasing visual-structural reasoning in molecular biology.

**Question**: What part is shown in this ultrasound image?
**Options**: A. Testicle, B. Large intestine, C. pancreas, D. Liver
**Answer**: C. pancreas.

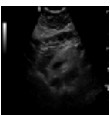

**Ours**: The ultrasound image shows a structure that appears to be a solid mass or lesion, which is consistent with a C. Plantar fascia pathology. This is typically seen in imaging of the foot and ankle region.
<answer>C</answer>

**Question**: The vestibular system can be stimulated or irritated by an infection of the inner ear or by a tumor (arrow) pressing on the vestibulocochlear nerve. Which of the following would NOT occur?
**Options**: A. Decreased auditory acuity. B. Vertigo. C. Tinnitis. D. Loss of facial sensation.
**Answer**: D. Loss of facial sensation
**Ours**: The image provided is an MRI scan of the brain, not of the inner ear or vestibular system. Therefore, it does not show any information related to the vestibular system or its components such as the vestibulocochlear nerve. The question asks about the vestibular system, which is not depicted in this image. Since the image does not contain relevant information for the question, we cannot determine what would or would not occur based on this image alone.
<answer>D. Loss of facial sensation.</answer>

Figure 8: Case 3 (Ultrasound image): The model interprets an ultrasound to identify the pancreas, integrating grayscale image features with organ shape and texture—an example of soft tissue recognition using sonography. Case 4 (Brain MRI with question about vestibular system): This task highlights the model's ability to identify mismatches between the image modality (brain MRI) and the clinical question (vestibular system), and to reason that facial sensation loss is not explainable from the given scan.

**Question**: Is polysplenia present?
**Options**: A. yes, B. no
**Answer**: A. yes.

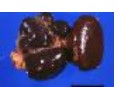

**Ours**: The image provided shows multiple organs that appear to be abnormally positioned or fused together, which is characteristic of polysplenia. Polysplenia is a condition where there are more than two spleens, often accompanied by other anomalies such as heart defects and other organ malformations.
<answer>A. yes</answer>

**Question**: Does malignant adenoma show bizarre strap and fusiform cells?
**Options**: A. yes, B. no
**Answer**: B. no

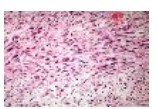

**Ours**: The image provided is a histological slide showing a pattern consistent with plantar fascia pathology. There is no indication of a malignant adenoma or bizarre strap and fusiform cells. Therefore, the correct answer is B.
<answer>B</answer>

Figure 9: Case 5 (Abdominal CT image – polysplenia diagnosis): The task involves recognizing abnormal organ positioning indicative of polysplenia. This case highlights anatomical reasoning through CT imaging, requiring recognition of fused or duplicated spleens and an understanding of associated syndromic features. Case 6 (Histopathology slide – adenoma diagnosis): A histological image is used to assess the presence of malignant features. The model correctly distinguishes normal plantar fascia morphology from pathological adenoma patterns, demonstrating reasoning in pathology image interpretation.

**Question**: Five elements make up 98% of the mass of the human body. These elements are oxygen (61%), carbon (23%), hydrogen (10.0%), nitrogen (2.6%), and calcium (1.4%). Compare these data with those in the pie graph below, which shows the five most abundant elements by mass in Earth's crust, oceans, and atmosphere. <image 1>,___,____,____,____,are abundant both in the human body and Earth's crust, oceans, and atmosphere? 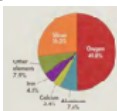

**Options**: A. iron, oxygen, Aluminum, silicon, B. Aluminum, silicon, iron, calcium, C. oxygen, silicon, iron, calcium

**Answer**: C. oxygen, silicon, iron, calcium.

**Ours**: The pie chart shows that the five most abundant elements by mass in Earth's crust, and atmosphere are oxygen (49.8%), silicon (26.0%), aluminum (7.6%), iron (4.8%), and calcium (3.4%). Comparing these with the five most abundant elements in the human body (oxygen, carbon, hydrogen, nitrogen, and calcium), we can see that oxygen is present in both sets of data.
<answer>C. oxygen, silicon, iron, calcium</answer>

Figure 10: Case 7 (Pie chart comparison – elemental composition): This case blends image data (a pie chart of Earth's element composition) with textual biochemical data (composition of the human body). The reasoning required crosses domains, comparing datasets to identify overlapping elements, exemplifying multimodal cross-referencing and synthesis.

