# OpenReview forum: "MMedAgent-RL: Optimizing Multi-Agent Collaboration for Multimodal Medical Reasoning"
_ICLR.cc/2026/Conference — ICLR 2026 Poster_

### Official Review · Reviewer_A9FJ · 2025-10-31

**Soundness:** 3
**Presentation:** 3
**Contribution:** 3
**Rating:** 6
**Confidence:** 4

**Summary:**

This paper proposes MMedAgent-RL, a multi-agent framework based on RL designed to optimize collaboration for multimodal medical diagnostic reasoning. The framework trains a triage doctor agent to assign cases and an attending physician agent, which uses a curriculum learning strategy, to integrate opinions from various specialist agents and make a final decision.

**Strengths:**

The proposed curriculum-based RL strategy to train an attending agent to evaluate and integrate potentially noisy or conflicting advice from specialists is a novel and well-motivated approach to improving the robustness of multi-agent systems.

**Weaknesses:**

My major comments are:

1. The specialist agents that provide the core domain knowledge are powerful proprietary models like GPT-4o and o3. This makes the framework's performance heavily dependent on external, closed-source models. The experiments do not clearly show how the system performs when only open-source models are used as specialists.
2.  The paper claims a significant performance gain over an SFT method. However, this SFT baseline is not clearly defined in the main text. It is uncertain if this is a model fine-tuned on ground-truth answers, on specialist responses, or another configuration.
3. The paper states the triage doctor is optimized using GRPO. However, the ablation study in Table 2 only presents a "w/o Triage" condition, which removes the step entirely. It fails to compare the RL-optimized triage agent against a simpler, non-RL baseline (e.g., a standard SFT-trained classifier). Given the near-perfect triage accuracy reported, the necessity of using RL for this component is not well-justified.
4.  The curriculum learning strategy splits training data based on specialist accuracy (easy, medium, hard). The paper then presents an analysis of performance on test data that is also split by difficulty. The method for splitting this test data is not explained. If it is based on the specialists' accuracy on the test set, this constitutes data leakage, as it uses information about the test answers to perform the analysis.

**Questions:**

Please see the weaknesses above.

---

> ### Author Response · Authors · 2025-11-21
> **Response to Reviewer A9FJ (1/2)**
>
> Thank you for your valuable feedback to help us improve our paper. We have revised our paper based on your feedback. We detail our response below and please kindly let us know if our response addresses your concerns.
>
> ****
>
> >**Q1**: The specialist agents that provide the core domain knowledge are powerful proprietary models like GPT-4o and o3. This makes the framework's performance heavily dependent on external, closed-source models. The experiments do not clearly show how the system performs when only open-source models are used as specialists.
>
> **A1**: We primarily chose OpenAI models as our specialist doctor due to its stable overall performance and fast response generation. A detailed analysis is provided in Section 5.3, "Analysis of Specialist Doctors," in our paper. As suggested, we have added additional ablation studies for various specialists using open-source VLMs, including HuatuoGPT-Vision and Qwen2.5-VL, as shown in **Table R1**. To demonstrate the generalization capability of our GP model, we do not re-train it to fit different specialists, but instead use the original GP model from our main experiments in paper. The results show that our method achieves strong performance with various combinations of specialists. The results show that our GP model maintains strong performance across various specialist configurations. While 3*o3 achieves the best overall results, combinations with HuatuoGPT-Vision and Qwen2.5-VL still perform competitively, demonstrating the model's robustness and flexibility.
> We have revised the paper in Section 5.3 and put these details in Appendix G.2.
>
> **Table R1**: Performance comparison with different specialists.
> | Model | VQA-RAD | SLAKE | PathVQA | OmniMedVQA | MMMU | Avg. |
> | :--- | :---: | :---: | :---: | :---: | :---: | :---: |
> | GPT-4o | 61.0 | 75.5 | 69.4 | 68.5 | 69.7 | 68.8 |
> | Qwen2.5-VL-7B | 61.8 | 64.7 | 60.5 | 60.8 | 56.6 | 60.9 |
> | HuatuoGPT-7B | 63.0 | 77.2 | 58.7 | 74.6 | 51.0 | 64.9 |
> | **MMedAgent-RL** | | | | | | |
> | 3\*OpenAI-o3 | 71.5 | 76.2 | 72.3 | 73.3 | 71.9 | 73.04 |
> | 3\*GPT-4o | 70.4 | 75.2 | 72.7 | 69.1 | 67.1 | 70.9 |
> | 1\*GPT-4o | 63.2 | 65.9 | 62.8 | 63.4 | 59.0 | 62.86 |
> | 1\*HuatuoGPT-Vision-7B | 63.4 | 67.2 | 60.8 | 64.9 | 54.1 | 62.1 |
> | 1\*Qwen2.5-VL-7B | 62.4 | 63.2 | 61.5 | 61.5 | 55.1 | 60.7 |
> | 2\*GPT-4o+1\*Qwen2.5-VL-7B | 70.0 | 74.6 | 71.3 | 66.0 | 64.0 | 69.2 |
> | 2\*GPT-4o+1\*HuatuoGPT-Vision-7B | 68.5 | 75.0 | 71.7 | 68.0 | 62.0 | 69.0 |
> | 2\*OpenAI-o3+1\*HuatuoGPT-Vision-7B | 69.9 | 75.8 | 73.0 | 70.0 | 68.0 | 71.3 |
> | 2\*OpenAI-o3+1\*Qwen2.5-VL-7B | 71.1 | 75.4 | 71.8 | 69.1 | 69.2 | 71.3 |
> | 1\*Qwen2.5-VL-7B+1\*GPT-4o | 68.9 | 73.8 | 70.6 | 66.1 | 63.2 | 68.5 |
> | 1\*HuatuoGPT-Vision-7B+1\*OpenAI-o3 | 69.0 | 74.7 | 72.3 | 68.8 | 68.3 | 70.6 |
> | 3\*HuatuoGPT-Vision-7B | 65.8 | 78.2 | 61.3 | 73.8 | 50.1 | 65.8 |
>
> ****
> >**Q2**: The paper claims a significant performance gain over an SFT method. However, this SFT baseline is not clearly defined in the main text. It is uncertain if this is a model fine-tuned on ground-truth answers, on specialist responses, or another configuration.
>
> **A2**: We apologize for the lack of clarity in the main text. To clarify, the SFT baseline refers to the General Practitioner (GP) agent trained via Supervised Fine-Tuning, using the same input context (including specialist knowledge) as our RL setting. We evaluated two specific configurations to ensure a fair comparison:
> - SFT w/o Reasoning Process: The model is fine-tuned directly on the ground-truth answers given the specialist context.
> - SFT w/ Reasoning Process: The model is fine-tuned on detailed reasoning traces distilled from a larger model (Qwen2.5-VL-32B).
>
> As shown in **Table R2**, while adding distilled reasoning traces improves performance over standard SFT (e.g., +3.3% on VQA-RAD), a significant gap remains compared to our method. MMedAgent-RL outperforms the best SFT baseline by a large margin (e.g., +11.3% on OmniMedVQA). This indicates that simple imitation (SFT) is insufficient for handling complex or conflicting expert advice. In contrast, our RL framework enables the agent to actively learn how to verify, weigh, and aggregate specialist information rather than merely memorizing the teacher's output.
> We have revised the paper and put these details in Appendix G.1.
>
> **Table R2**: Performance comparison with SFT-based methods.
> | Method | VQA-RAD | SLAKE | PathVQA | OmniMedVQA | MMMU-Med | Overall |
> | :--- | :---: | :---: | :---: | :---: | :---: | :---: |
> | Qwen2.5-VL-7B | 61.8 | 64.7 | 60.5 | 60.8 | 56.6 | 60.9 |
> | + SFT w/o reasoning process | 65.5 | 66.5 | 61.4 | 60.9 | 57.8 | 62.4 |
> | + SFT w/ reasoning process | 68.8 | 68.4 | 63.7 | 62.0 | 59.4 | 64.5 |
> | **MMedAgent-RL** | **71.5** | **76.2** | **72.3** | **73.3** | **71.9** | **73.0** |

---

> > ### Author Response · Authors · 2025-11-21
> > **Response to Reviewer A9FJ (2/2)**
> >
> > >**Q3**: The paper states the triage doctor is optimized using GRPO. However, the ablation study in Table 2 only presents a "w/o Triage" condition, which removes the step entirely. It fails to compare the RL-optimized triage agent against a simpler, non-RL baseline (e.g., a standard SFT-trained classifier). Given the near-perfect triage accuracy reported, the necessity of using RL for this component is not well-justified.
> >
> > **A3**: We appreciate this insightful comment. You are correct that the original "w/o Triage" ablation was insufficient to justify the specific choice of RL over simpler methods. To address this, we conducted a detailed comparison between our GRPO-optimized Triage Agent and standard Supervised Fine-Tuning (SFT) baselines.
> > As shown in **Table R3**, we evaluated two SFT configurations:
> > - SFT (w/o Reasoning): Fine-tuned on direct Question-Department pairs.
> > - SFT (w/ Reasoning): Fine-tuned using reasoning traces distilled from Qwen2.5-VL-32B.
> >
> > While SFT significantly improves performance over the base model, GRPO still outperforms the best SFT baseline across all datasets. Crucially, this advantage is most pronounced on Out-of-Distribution (OOD) datasets. For instance, on MMMU-Med, GRPO outperforms "SFT w/ Reasoning" by +4.1% (71.9 vs. 67.8), and on OmniMedVQA by +2.5% (73.3 vs. 70.8).
> > This indicates that while SFT can achieve high accuracy on standard distributions, the reinforcement learning process (GRPO) enables the Triage Agent to generalize better to complex, unseen scenarios by learning from the reward signal of the downstream reasoning success, rather than just mimicking a static label.
> > We have revised the paper and put these details in Appendix G.6.
> >
> > **Table R3**: Performance comparison of Triage Agent training strategies.
> > | Method | VQA-RAD | SLAKE | PathVQA | OmniMedVQA | MMMU-Med |
> > | :--- | :---: | :---: | :---: | :---: | :---: |
> > | Base model | 66.3 | 69.9 | 67.2 | 66.2 | 59.3 |
> > | + SFT w/o reasoning process | 69.4 | 75.0 | 70.5 | 70.1 | 66.7 |
> > | + SFT w/ reasoning process | 70.2 | 75.9 | 71.0 | 70.8 | 67.8 |
> > | + GRPO | **71.5** | **76.2** | **72.3** | **73.3** | **71.9** |
> >
> >
> > ****
> >
> > >**Q4**: The curriculum learning strategy splits training data based on specialist accuracy (easy, medium, hard). The paper then presents an analysis of performance on test data that is also split by difficulty. The method for splitting this test data is not explained. If it is based on the specialists' accuracy on the test set, this constitutes data leakage, as it uses information about the test answers to perform the analysis.
> >
> > **A4**: We respectfully clarify that there is no data leakage.
> > The partition of training, validation, and test sets strictly follows the official dataset guidelines.
> > The difficulty grading is required exclusively for the training phase to implement curriculum learning. Specifically, we classify samples into difficulty levels based on the consistency and accuracy of the specialists' responses (as detailed in Section 3.3).
> > We emphasize that the inference process and the calculation of evaluation scores do not use this difficulty grading at all. The model generates responses without access to any difficulty labels or ground truth.
> > The difficulty stratification on the test set is applied solely for the analytical breakdown shown in Figure 4. It serves only to visualize and analyze model performance across different complexity levels post-generation, and has no influence on the inference process itself. The difficulty grading for the test dataset follows the same as training data. We have revised the paper in Section 5.1 and put these details in Appendix F.4.

---

> ### Comment · Reviewer_A9FJ · 2025-11-26
>
> Thanks to the authors for their response. Most of my questions have been addressed. I'll keep my score.

---

### Official Review · Reviewer_dwrJ · 2025-10-31

**Soundness:** 4
**Presentation:** 4
**Contribution:** 4
**Rating:** 8
**Confidence:** 4

**Summary:**

This paper develops a RL-based framework, namely MMedAgent-RL, for optimizing multi-agent collaboration in medical VLM reasoning. The proposed framework is well-motivated and empirically strong, with evaluations on both in-domain and out-of-distribution datasets.

**Strengths:**

(1) This framework develops a machine that can adjust collaboration policies based on task difficulty. The integration of C-MARL for entropy control is theoretically motivated.
(2) The evaluation and experimental design are comprehensive. The proposed framework shows good performance in 5 public datasets, including both in-domain and out-of-distribution datasets.

**Weaknesses:**

(1) The framework is inspired by the 'triage–specialist–attending'. The authors need to find more evidence to demonstrate that this aligns with the real hospitalization process. Within different sections in a hospital, the workflow may differ.
(2) This work lacks the involvement of human experts.
(3) Some of the technical details are missing. For example, are the first GP and the second GP updated simultaneously?

**Questions:**

The authors use Qwen2.5-VL as the base model. Is this framework transferable to other base models?

---

> ### Author Response · Authors · 2025-11-21
> **Response to Reviewer dwrJ (1/2)**
>
> Thank you for your constructive comments and suggestions. We have revised our paper according to your comments. We respond to your questions below and would appreciate it if you could let us know if our response addresses your concerns.
>
> ****
>
> >**Q1**: The framework is inspired by the 'triage–specialist–attending'. The authors need to find more evidence to demonstrate that this aligns with the real hospitalization process. Within different sections in a hospital, the workflow may differ.
>
> **A1**: We acknowledge that specific clinical workflows vary across hospital departments. However, our framework abstracts the fundamental Medical Consultation and Multi-Disciplinary Team process, which is the standard standard for handling complex diagnoses: initial assessment (triage), expert consultation (specialists), and final synthesis (attending physician).
> This collaborative paradigm is consistent with established methodologies in recent medical AI literature. Previous works such as MedAgents [1] and MDAgents [2] have demonstrated that modeling such role-based, multi-round collaboration effectively simulates real-world medical decision-making and significantly enhances diagnostic reasoning. Our framework builds upon this proven abstraction to optimize information routing and aggregation.
> We have revised the paper in Section 3.1.
>
> [1] MedAgents: Large Language Models as Collaborators for Zero-shot Medical Reasoning. ACL 2024 Findings.
>
> [2] MDAgents: An Adaptive Collaboration of LLMs for Medical Decision-Making. NeurIPS 2024.
>
> ****
>
> >**Q2**: This work lacks the involvement of human experts.
>
> **A2**: Human expert involvement was a core component of our evaluation methodology, serving both as a performance benchmark and a qualitative evaluator. To ensure clinical relevance, we engaged three practicing clinical experts to evaluate 50 randomly selected samples. This study focused on two key dimensions:
>
> Diagnostic Accuracy Benchmark: The experts provided their own diagnoses for the samples to establish a "Human Upper Bound." As shown in **Table R1**, while human experts achieved 98.0% accuracy, our MMedAgent-RL reached 82.0%, significantly narrowing the gap compared to standard baselines (e.g., LLaVA-Med at 44.0%).
>
> Reasoning Process Evaluation: The experts also scored the quality of the models' reasoning chains on a 1–5 scale (normalized in the table). They assessed whether the models followed a logical clinical workflow (e.g., defining the disease $\rightarrow$ analyzing the image $\rightarrow$ consistency checking). As shown in **Table R1**, the experts confirmed that our method produces reasoning paths that align more closely with clinical standards, achieving a score of 72.0, compared to only 28.0 for LLaVA-Med.
> We have revised the manuscript in Appendix G.9 to make this expert evaluation more prominent.
>
> **Table R1**: Human Expert Assessment
>
> | Model | Acc | Reasoning Score (by human) |
> | :--- | :---: | :---: |
> | Human | 98.0 | - |
> | Qwen2.5-VL-3B | 62.0 | 38.0 |
> | LLaVA-Med-7B | 44.0 | 28.0 |
> | **MMedAgent-RL-7B** | **82.0** | **72.0** |
>
>
> ****
> >**Q3**:  Some of the technical details are missing. For example, are the first GP and the second GP updated simultaneously?
>
> **A3**: Thank you for noting this. In the paper, the two GP agents (the Triage Agent and the Attending Physician) are updated independently.
> This design choice was primarily made to ensure training stability and to decouple the training process from potential API failures or latency when querying the external OpenAI-based specialists. Specifically, our procedure is as follows:
>
> - Triage Optimization: We first optimize the Triage Agent using image-modality QA pairs to ensure accurate department routing.
> - Data Preparation: We classify the training data based on these departments and invoke the OpenAI API (acting as specialists) to generate expert knowledge offline.
> - Attending GP Training: Finally, we use these pre-generated expert trajectories to train the Attending GP (Qwen2.5-VL) via Reinforcement Learning.
>
> However, following your suggestion, we also implemented an end-to-end online framework where both GPs are updated simultaneously. As shown in **Table R2**, the performance difference between the two settings is negligible. This confirms that our decoupled training strategy is valid and yields results consistent with a fully end-to-end approach while remaining more computationally efficient and stable.
> We have revised the paper and put these details in Appendix G.10.
>
> **Table R2**: Performance comparison of GP update strategies.
> | Strategy | VQA-RAD | SLAKE | PathVQA | OmniMedVQA | MMMU-Med | Overall |
> | :--- | :---: | :---: | :---: | :---: | :---: | :---: |
> | Independent | **71.5** | 76.2 | 72.3 | 73.3 | **71.9** | 73.0 |
> | Simultaneous | 71.3 | **76.5** | **72.4** | **73.6** | 71.6 | **73.1** |

---

> > ### Author Response · Authors · 2025-11-21
> > **Response to Reviewer dwrJ (2/2)**
> >
> > >**Q4**: The authors use Qwen2.5-VL as the base model. Is this framework transferable to other base models?
> >
> > **A4**: To demonstrate the transferability of our framework, we conducted additional experiments using InternVL2.5-Instruct-8B as an alternative base model.
> > As shown in **Table R3**, our method yields consistent and significant improvements across all datasets, regardless of the backbone architecture. On OmniMedVQA, the InternVL-based agent achieved a remarkable score of 82.4%, surpassing the performance of the Qwen-based version. Even on datasets where the base InternVL model struggled (e.g., PathVQA, where the base score was only 42.3%), our framework provided a massive performance boost of +26.1% (reaching 68.4%). These results confirm that our pipeline is model-agnostic and can effectively enhance the reasoning capabilities of diverse multimodal LLMs.
> > We have revised the paper and put these details in Appendix G.11.
> >
> > **Table R3**: Assessment of framework transferability.
> > | Model | VQA-RAD | SLAKE | PathVQA | OmniMedVQA | MMMU-Med |
> > | :--- | :---: | :---: | :---: | :---: | :---: |
> > | Qwen2.5-VL-7B | 61.8 | 64.7 | 60.5 | 60.8 | 56.6 |
> > | MMedAgent-RL (Qwen2.5-VL-7B) | **71.5** | **76.2** | **72.3** | **73.3** | **71.9** |
> > | InternVL2.5-8B | 58.6 | 68.6 | 42.3 | 76.5 | 51.4 |
> > | MMedAgent-RL (InternVL2.5-8B) | **70.2** | **78.9** | **68.4** | **82.4** | **64.7** |

---

### Official Review · Reviewer_noLG · 2025-10-31

**Soundness:** 2
**Presentation:** 2
**Contribution:** 2
**Rating:** 4
**Confidence:** 3

**Summary:**

This paper introduces MMedAgent-RL, a reinforcement learning (RL)-based multi-agent framework that overcomes the rigidity of existing collaboration systems by enabling dynamic and optimized cooperation among medical agents. The framework mimics a clinical "triage-and-referral" system, utilizing a curriculum RL strategy with dynamic entropy regulation to train a primary model to intelligently integrate and resolve noisy or conflicting inputs from various specialist agents.

**Strengths:**

1. The paper is well-written, logically clear, and easy to follow.
2. The theoretical derivations are fairly sound.
3. Extensive experiments demonstrate the superiority of the proposed MMedAgent-RL.

**Weaknesses:**

1. The middle part of Figure 1(a) does not reflect the practical workflow of Multi-Agent collaboration; it seems to lack representation of the General Practitioner, which leads to ambiguity.
2. Section 3.1 mentions optimizing the triage doctor using GRPO, so it would be worthwhile to discuss the triage doctor's capability (quantitatively) as well as its reasoning process.
3. The underlying mechanism for the entropy regularization term in Equation 3.1 needs to be explained, and the rationale behind the choice and range of values for $\gamma_s$ should also be elaborated.
4. Figure 3 mentions the selection of three 'o3' models as specialist doctors? Does this mean each specialty uses three 'o3's? If so, there seems to be no differentiation between the specialties. Why was 'o3' chosen over a specially designed medical MLLM?
5. The experimental setup in Figure 4 is not clearly described, and more details should be provided.

**Questions:**

Please refer to Weaknesses.

---

> ### Author Response · Authors · 2025-11-21
> **Response to Reviewer noLG (1/2)**
>
> Thank you for reviewing our paper and for your valuable feedback. Below, we address your concerns point by point and we’ve revised our paper according to your suggestions. We would appreciate it if you could let us know whether your concerns are addressed by our response.
>
> ****
>
> >**Q1**: The middle part of Figure 1(a) does not reflect the practical workflow of Multi-Agent collaboration; it seems to lack representation of the General Practitioner, which leads to ambiguity.
>
> **A1**: Thank you for pointing this out. We have revised Figure 1 to explicitly include the General Practitioner and the multi-agent collaboration module to avoid any ambiguity.
> To clarify our original intent: the middle section of Figure 1(a) was initially designed to illustrate the challenge that standard multi-agent collaboration workflows are difficult to optimize directly. However, we agree that the previous visualization was unclear. The updated figure now accurately reflects the practical workflow.
>
> ****
>
> >**Q2**: Section 3.1 mentions optimizing the triage doctor using GRPO, so it would be worthwhile to discuss the triage doctor's capability (quantitatively) as well as its reasoning process.
>
> **A2**: We appreciate this valuable suggestion regarding the Triage Doctor's optimization. As shown in Table R1, we quantitatively evaluated the impact of different training stages on the triage agent’s performance.
> We compared the Base Model, SFT (trained on direct question-answer pairs without reasoning), SFT with Reasoning (trained on reasoning traces distilled from Qwen2.5-VL-32B), and our final GRPO-optimized model. The results demonstrate that the triage agent trained with GRPO yields the highest performance. Notably, this improvement is most significant on the two Out-of-Distribution (OOD) datasets, i.e., OmniMedVQA and MMMU-Med, confirming that the reasoning capabilities reinforced by GRPO are crucial for generalization.
> We have revised the paper and put these details in Appendix G.8.
>
> **Table R1**: Performance with triage agent with different settings.
> | Method | VQA-RAD | SLAKE | PathVQA | OmniMedVQA | MMMU-Med | Overall |
> | :--- | :---: | :---: | :---: | :---: | :---: | :---: |
> | Base model | 66.3 | 69.9 | 67.2 | 66.2 | 59.3 | 65.8 |
> | + SFT w/o reasoning process | 69.4 | 75.0 | 70.5 | 70.1 | 66.7 | 70.3 |
> | + SFT w/ reasoning process | 70.2 | 75.9 | 71.0 | 70.8 | 67.8 | 71.1 |
> | + GRPO | **71.5** | **76.2** | **72.3** | **73.3** | **71.9** | **73.0** |
>
> ****
>
> >**Q3**: The underlying mechanism for the entropy regularization term in Equation 3.1 needs to be explained, and the rationale behind the choice and range of values for
>  should also be elaborated.
>
> **A3**: The entropy term $H_t$ serves as an intrinsic exploration bonus to prevent the policy from becoming overly deterministic. The coefficient $\gamma_s$ is dynamically adjusted based on the curriculum difficulty ($s$) to balance exploration and exploitation:
>
> * Reliable ($s=1$): We set $\gamma \approx 0$ to encourage *exploitation* of the trustworthy specialist consensus.
> * Conflicting ($0 < s < 1$): We use a moderate $\gamma > 0$ to prevent overconfidence when specialists disagree.
> * Misleading ($s=0$): We apply a large $\gamma \gg 0$ to enforce *aggressive exploration*, compelling the model to deviate from the incorrect specialist consensus.
> We have revised the paper and highlighted these in Section 3.3.

---

> > ### Author Response · Authors · 2025-11-21
> > **Response to Reviewer noLG (2/2)**
> >
> > >**Q4**: Figure 3 mentions the selection of three 'o3' models as specialist doctors? Does this mean each specialty uses three 'o3's? If so, there seems to be no differentiation between the specialties. Why was 'o3' chosen over a specially designed medical MLLM?
> >
> > **A4**: You are correct that Figure 3 utilizes three ‘o3’ models as specialist agents. However, Specialists are defined differently based on the prompts that specify distinct medical departments. Consequently, even though the underlying model is the same, the system prompts and resulting perspectives differ significantly. Furthermore, multiple sampling introduces valuable diversity that aids the reasoning process.
> >
> > Regarding the choice of model, we initially selected OpenAI models for its superior stability and inference efficiency. However, following your suggestion, we have conducted additional ablation studies using open-source VLMs, specifically HuatuoGPT-Vision (a medical-specific MLLM) and Qwen2.5-VL, to validate our framework's flexibility. The results are presented in **Table R2**.
> > Crucially, to demonstrate the generalization capability of our General Practitioner (GP) agent, we did not retrain the GP to fit these new specialists. Instead, we deployed the original GP model trained in our main experiments. As shown in **Table R2**, while the 3 × o3 configuration achieves the highest performance, our framework remains highly effective with varying combinations of open-source specialists, demonstrating the GP's robustness in coordinating diverse expert models.
> > We have revised the paper in Section 5.3 and put these details in Appendix G.2.
> >
> >
> > **Table R2**: Performance comparison with different specialists.
> > | Model | VQA-RAD | SLAKE | PathVQA | OmniMedVQA | MMMU | Avg. |
> > | :--- | :---: | :---: | :---: | :---: | :---: | :---: |
> > | GPT-4o | 61.0 | 75.5 | 69.4 | 68.5 | 69.7 | 68.8 |
> > | Qwen2.5-VL-7B | 61.8 | 64.7 | 60.5 | 60.8 | 56.6 | 60.9 |
> > | HuatuoGPT-7B | 63.0 | 77.2 | 58.7 | 74.6 | 51.0 | 64.9 |
> > | **MMedAgent-RL** | | | | | | |
> > | 3\*OpenAI-o3 | 71.5 | 76.2 | 72.3 | 73.3 | 71.9 | 73.04 |
> > | 3\*GPT-4o | 70.4 | 75.2 | 72.7 | 69.1 | 67.1 | 70.9 |
> > | 1\*GPT-4o | 63.2 | 65.9 | 62.8 | 63.4 | 59.0 | 62.86 |
> > | 1\*HuatuoGPT-Vision-7B | 63.4 | 67.2 | 60.8 | 64.9 | 54.1 | 62.1 |
> > | 1\*Qwen2.5-VL-7B | 62.4 | 63.2 | 61.5 | 61.5 | 55.1 | 60.7 |
> > | 2\*GPT-4o+1\*Qwen2.5-VL-7B | 70.0 | 74.6 | 71.3 | 66.0 | 64.0 | 69.2 |
> > | 2\*GPT-4o+1\*HuatuoGPT-Vision-7B | 68.5 | 75.0 | 71.7 | 68.0 | 62.0 | 69.0 |
> > | 2\*OpenAI-o3+1\*HuatuoGPT-Vision-7B | 69.9 | 75.8 | 73.0 | 70.0 | 68.0 | 71.3 |
> > | 2\*OpenAI-o3+1\*Qwen2.5-VL-7B | 71.1 | 75.4 | 71.8 | 69.1 | 69.2 | 71.3 |
> > | 1\*Qwen2.5-VL-7B+1\*GPT-4o | 68.9 | 73.8 | 70.6 | 66.1 | 63.2 | 68.5 |
> > | 1\*HuatuoGPT-Vision-7B+1\*OpenAI-o3 | 69.0 | 74.7 | 72.3 | 68.8 | 68.3 | 70.6 |
> > | 3\*HuatuoGPT-Vision-7B | 65.8 | 78.2 | 61.3 | 73.8 | 50.1 | 65.8 |
> >
> > ****
> >
> > >**Q5**: The experimental setup in Figure 4 is not clearly described, and more details should be provided.
> >
> > **A5**: We apologize for the ambiguity regarding the experimental setup in Figure 4. We have revised the manuscript to provide a detailed definition of the "Decision Difficulty" levels and the model configurations.
> > To clarify here: Figure 4 evaluates the robustness of the General Practitioner (GP) agent against noisy or misleading specialist advice. The setup is defined as follows:
> > - Difficulty Levels (X-axis): We categorize test samples based on the consistency and accuracy of the specialists' advice:
> > - Easy: All specialists provide correct and consistent answers.
> > - Medium: Specialists provide conflicting advice (disagreement).
> > - Hard: Specialists consistently provide incorrect advice (misleading consensus).
> > - Model Variants (Legend): We compare the progression of the Qwen2.5-VL based GP agent:
> > - Red/Pink bars: The baseline performance (Base model and SFT) without our reinforcement learning framework.
> > - Blue/Yellow bars (C-MARL): The performance of our method at different stages of the curriculum training.
> >
> > As observed in the figure, since specialists cannot guarantee complete accuracy, "Hard" samples (noise) significantly impact the baseline models. However, our C-MARL method enables the agent to gradually learn to distinguish and utilize specialist knowledge effectively. As a result, the overall performance is 20% higher than the original model, with the most significant gains observed in the "Hard" category, demonstrating the agent's ability to overcome misleading specialist hallucinations.
> > We have revised the paper in Section 5.3.

---

> > > ### Author Response · Authors · 2025-11-26
> > >
> > > Dear reviewer noLG,
> > >
> > > We would like to follow up to see if the response addresses your concerns. We would really appreciate the opportunity to discuss this further if our response has not already addressed your concerns. Thank you again!

---

### Official Review · Reviewer_NzMH · 2025-11-03

**Soundness:** 2
**Presentation:** 3
**Contribution:** 2
**Rating:** 4
**Confidence:** 3

**Summary:**

This paper proposes MMedAgent-RL, a reinforcement learning framework for multi-agent medical reasoning. The system simulates clinical workflows with a triage doctor routing cases to specialists (proprietary LVLMs), then an attending physician trained via curriculum RL to aggregate specialist opinions. The key innovation is a curriculum learning strategy with dynamic entropy regulation that progressively teaches the model to handle specialist outputs of varying reliability. Experiments on 5 medical VQA benchmarks show significant gains over baselines.

**Strengths:**

1. Framing multi-agent medical reasoning as a curriculum RL problem with dynamic entropy control is well-motivated by the reality of imperfect expert judgments.

2. Strong empirical results, 23.6% average gain over baselines and excellent OOD generalization (72.6% on MMMU/OmniMedVQA) demonstrate effectiveness.

3. The three-stage curriculum (easy/medium/hard based on specialist accuracy) with corresponding entropy coefficients (0.0001/0.005/0.03) is principled and clearly explained.

**Weaknesses:**

1. Missing critical baselines: No comparison with simpler alternatives that could  possible achieve similar results, eg. single GPT-4o or Qwen2.5-VL sampling N diverse outputs using different prompts or high temperatures → majority voting or trained aggregator. These would test if the complex triage+multi-expert pipeline is necessary.

2. The paper claims the attending physician learns to "correct specialist mistakes," but provides no quantitative evidence on hard cases where all specialists fail. A fairer and more rigorous evaluation is needed to determine how much of the performance gain is due to routing and how much is due to aggregation.

3. Best performance requires OpenAI llms, but medical data cannot be sent to external APIs in privacy-critical environments. Where is the evaluation with deployable open-source specialists ?

I will reconsider my rating if the author addresses these questions well.

**Questions:**

1. In "w/o Triage" (Table 2), what exactly happens? Random specialist? All specialists? Please clarify and consider: single model with diverse sampling might achieve similar diversity without routing.

2. Can you provide results progressively adding components (base → +triage → +multi-expert → +curriculum RL) to quantify each contribution?

---

> ### Author Response · Authors · 2025-11-21
> **Response to Reviewer NzMH (1/3)**
>
> Thank you for your valuable feedback to help us improve our paper. We have revised our paper based on your feedback. We detail our response below and please kindly let us know if our response addresses your concerns.
>
> ****
>
> >**Q1**: Missing critical baselines: No comparison with simpler alternatives that could possible achieve similar results, e.g. single GPT-4o or Qwen2.5-VL sampling N diverse outputs using different prompts or high temperatures → majority voting or trained aggregator. These would test if the complex triage+multi-expert pipeline is necessary.
>
> **A1**: We conducted additional experiments using test-time scaling techniques, specifically Majority Voting and Self-Consistency [1], on both Qwen2.5-VL-7B and GPT-4o. For the experimental setup, Majority Voting involved sampling $N=3$ outputs and selecting the most frequent answer as the final prediction. Self-Consistency [1] sampled a set of diverse reasoning paths rather than relying on the greedy path, subsequently selecting the most consistent answer by marginalizing over the sampled paths. As shown in **Table R1**, test-time scaling improved the performance of Qwen2.5-VL-7B by 2.6% and GPT-4o by 1.6% across the five datasets. However, despite these improvements, a performance gap remains compared to our multi-agent framework. Our method outperforms these enhanced baselines by 18%, demonstrating the necessity and effectiveness of our proposed triage and multi-expert pipeline. We have revised the paper and put these details in Appendix G.4.
>
>
> **Table R1**: Performance comparison with test-time scaling baselines.
> | Method | VQA-RAD | SLAKE | PathVQA | OmniMedVQA | MMMU-Med | Overall |
> | :--- | :---: | :---: | :---: | :---: | :---: | :---: |
> | Qwen2.5-VL-7B | 61.8 | 64.7 | 60.5 | 60.8 | 56.6 | 60.9 |
> | Qwen2.5-VL-7B + Majority Voting | 63.7 | 65.4 | 61.5 | 62.5 | 57.0 | 62.0 |
> | Qwen2.5-VL-7B + Self-consistency | 63.8 | 65.3 | 62.4 | 64.5 | 59.1 | 63.0 |
> | GPT-4o | 61.0 | 75.5 | 69.4 | 68.5 | 69.7 | 68.8 |
> | GPT-4o + Majority Voting | 62.2 | 75.9 | 70.3 | 69.4 | 70.3 | 69.6 |
> | GPT-4o + Self-consistency | 63.6 | 75.4 | 70.8 | 69.1 | 71.5 | 70.1 |
> | **MMedAgent-RL** | **71.5** | **76.2** | **72.3** | **73.3** | **71.9** | **73.0** |
>
> [1] Wang, Xuezhi, et al. "Self-Consistency Improves Chain of Thought Reasoning in Language Models." ICLR 2023.
>
> ****
>
> >**Q2**: The paper claims the attending physician learns to "correct specialist mistakes," but provides no quantitative evidence on hard cases where all specialists fail. A fairer and more rigorous evaluation is needed to determine how much of the performance gain is due to routing and how much is due to aggregation.
>
> **A2**: To strictly quantify the model's ability to correct mistakes rather than simply route to a specialist, we conducted a targeted evaluation on hard samples where all three specialists provided incorrect answers.
> We selected 200 such samples from PathVQA and 200 from OmniMedVQA to evaluate the performance of the Base Model (with and without expert knowledge), the SFT baseline, and our MMedAgent-RL.
> Since every specialist is incorrect, any routing mechanism would yield 0.0% accuracy. Therefore, the performance on this subset is strictly attributable to the model's ability to perform intrinsic correction. As shown in **Table R2**, our method achieves substantial improvements. For instance, on the OOD OmniMedVQA hard subset, MMedAgent-RL improves accuracy to 23.0%.
> We attribute this capability to the following logic:
> Inability to self-answer: The extremely low accuracy of the Base Model w/o expert knowledge (4.5% on PathVQA and 2.0% on OmniMedVQA) confirms that the model lacks the intrinsic parametric knowledge to solve these hard cases independently.
>
> Reasoning extraction over blind trust: Although the specialists' final answers were wrong, their reasoning processes likely contained partial truths or valid medical context. While the SFT baseline struggles to utilize this conflicting information (often hallucinating along with the experts), MMedAgent-RL has learned via RL to critically synthesize these valid reasoning fragments, correcting the final conclusion rather than simply aggregating the errors.
> We have revised the paper and put these details in Appendix G.5.
>
> **Table R2**: Correctness ratio (accuracy on hard samples where all specialists failed).
> | Setting / Model | PathVQA | OmniMedVQA |
> | :--- | :---: | :---: |
> | **w/o expert knowledge** | | |
> | &nbsp;&nbsp;&nbsp;&nbsp;Base Model | 4.5% | 2.0% |
> | **w/ expert knowledge** | | |
> | &nbsp;&nbsp;&nbsp;&nbsp;Base Model | 5.5% | 3.5% |
> | &nbsp;&nbsp;&nbsp;&nbsp;SFT | 10.0% | 7.5% |
> | &nbsp;&nbsp;&nbsp;&nbsp;**MMedAgent-RL** | **26.5%** | **23.0%** |

---

> > ### Author Response · Authors · 2025-11-21
> > **Response to Reviewer NzMH (2/3)**
> >
> > >**Q3**: Best performance requires OpenAI llms, but medical data cannot be sent to external APIs in privacy-critical environments. Where is the evaluation with deployable open-source specialists?
> >
> > **A3**: We primarily chose OpenAI models as our specialist doctor due to its stable overall performance and fast response generation. The detailed analysis was provided in Section 5.3, "Analysis of Specialist Doctors," in our paper. As suggested, we have added additional ablation studies for various specialists using open-source VLMs, including HuatuoGPT-Vision and Qwen2.5-VL, as shown in **Table R3**. To demonstrate the generalization capability of our GP model, we do not re-train it to fit different specialists, but instead use the original GP model from our main experiments in paper. The results show that our method achieves strong performance with various combinations of specialists. While 3*o3 achieves the best overall results, combinations with HuatuoGPT-Vision and Qwen2.5-VL still perform competitively, demonstrating the model's robustness and flexibility.
> > We have revised the paper in Section 5.3 and put these details in Appendix G.2.
> >
> > **Table R3**: Performance comparison with different specialists.
> > | Model | VQA-RAD | SLAKE | PathVQA | OmniMedVQA | MMMU | Avg. |
> > | :--- | :---: | :---: | :---: | :---: | :---: | :---: |
> > | GPT-4o | 61.0 | 75.5 | 69.4 | 68.5 | 69.7 | 68.8 |
> > | Qwen2.5-VL-7B | 61.8 | 64.7 | 60.5 | 60.8 | 56.6 | 60.9 |
> > | HuatuoGPT-7B | 63.0 | 77.2 | 58.7 | 74.6 | 51.0 | 64.9 |
> > | **MMedAgent-RL** | | | | | | |
> > | 3\*OpenAI-o3 | 71.5 | 76.2 | 72.3 | 73.3 | 71.9 | 73.04 |
> > | 3\*GPT-4o | 70.4 | 75.2 | 72.7 | 69.1 | 67.1 | 70.9 |
> > | 1\*GPT-4o | 63.2 | 65.9 | 62.8 | 63.4 | 59.0 | 62.86 |
> > | 1\*HuatuoGPT-Vision-7B | 63.4 | 67.2 | 60.8 | 64.9 | 54.1 | 62.1 |
> > | 1\*Qwen2.5-VL-7B | 62.4 | 63.2 | 61.5 | 61.5 | 55.1 | 60.7 |
> > | 2\*GPT-4o+1\*Qwen2.5-VL-7B | 70.0 | 74.6 | 71.3 | 66.0 | 64.0 | 69.2 |
> > | 2\*GPT-4o+1\*HuatuoGPT-Vision-7B | 68.5 | 75.0 | 71.7 | 68.0 | 62.0 | 69.0 |
> > | 2\*OpenAI-o3+1\*HuatuoGPT-Vision-7B | 69.9 | 75.8 | 73.0 | 70.0 | 68.0 | 71.3 |
> > | 2\*OpenAI-o3+1\*Qwen2.5-VL-7B | 71.1 | 75.4 | 71.8 | 69.1 | 69.2 | 71.3 |
> > | 1\*Qwen2.5-VL-7B+1\*GPT-4o | 68.9 | 73.8 | 70.6 | 66.1 | 63.2 | 68.5 |
> > | 1\*HuatuoGPT-Vision-7B+1\*OpenAI-o3 | 69.0 | 74.7 | 72.3 | 68.8 | 68.3 | 70.6 |
> > | 3\*HuatuoGPT-Vision-7B | 65.8 | 78.2 | 61.3 | 73.8 | 50.1 | 65.8 |
> >
> > ****
> >
> > >**Q4**: In "w/o Triage" (Table 2), what exactly happens? Random specialist? All specialists? Please clarify and consider: single model with diverse sampling might achieve similar diversity without routing.
> >
> > **A4**: In Table 2, "w/o Triage" refers to using the original base model to perform the routing (triage) task directly, without specific fine-tuning for this role. Following your suggestions, we have expanded our comparison in **Table R4**. We introduced two new settings to test the necessity of the framework:
> >
> > - Random Routing: Assigning queries to specialists randomly to isolate the benefit of the specialists themselves.
> > - Single Model w/o Routing: Using the Qwen2.5-VL-7B base model directly, and enhancing it with Majority Voting (diverse sampling) as requested.
> >
> > As shown in **Table R4**, incorporating a routing mechanism leads to significant performance gains. Notably, even Random Routing generally outperforms the single model equipped with Majority Voting. Furthermore, our proposed MMedAgent-RL significantly outperforms all baselines, confirming that a dedicated triage-and-expert pipeline provides advantages that cannot be achieved by simple diverse sampling or random assignment.
> > We have revised the paper and put these details in Appendix G.6.
> >
> > **Table R4**: Performance comparison with or without routing.
> > | Method | VQA-RAD | SLAKE | PathVQA | OmniMedVQA | MMMU-Med | Overall |
> > | :--- | :---: | :---: | :---: | :---: | :---: | :---: |
> > | **w/ triage** | | | | | | |
> > | &nbsp;&nbsp;&nbsp;&nbsp;MMedAgent-RL (zero-shot) | 66.3 | 69.9 | 67.2 | 66.2 | 59.3 | 65.8 |
> > | &nbsp;&nbsp;&nbsp;&nbsp;MMedAgent (random triage) | 63.4 | 66.7 | 64.7 | 67.8 | 58.2 | 64.2 |
> > | &nbsp;&nbsp;&nbsp;&nbsp;**MMedAgent-RL (fine-tuned triage)** | **71.5** | **76.2** | **72.3** | **73.3** | **71.9** | **73.0** |
> > | **w/o triage** | | | | | | |
> > | &nbsp;&nbsp;&nbsp;&nbsp;Qwen2.5-VL-7B | 61.8 | 64.7 | 60.5 | 60.8 | 56.6 | 60.9 |
> > | &nbsp;&nbsp;&nbsp;&nbsp;+ Majority Voting | 63.7 | 65.4 | 61.5 | 62.5 | 57.0 | 62.0 |

---

> > ### Comment · Reviewer_NzMH · 2025-11-25
> >
> > The author may have misunderstood the key baseline I proposed.
> >
> > I believe the most important baseline is to utilize GPT-4o to sample N diverse outputs using different prompts or high temperatures, and with a trained aggregator (not majority voting or self-consistency).
> >
> > If this works, does it mean we no longer need cumbersome routing?

---

> > > ### Author Response · Authors · 2025-11-26
> > >
> > > We sincerely appreciate the suggestion to compare against a stronger "Generator + Trained Aggregator" baseline. To rigorously address this, we implemented two new baselines where GPT-4o first samples $N=3$ diverse outputs per query ($T=1.0$), and a Qwen2.5-VL-7B model is then trained on the training set (same as data we used) to act as an aggregator that selects or synthesizes the final answer from these candidates. We developed both an SFT Aggregator (via Supervised Fine-Tuning) and a GRPO Aggregator (via Group Relative Policy Optimization) to ensure a robust comparison. As shown in **Table R1**, MMedAgent-RL still maintains a significant performance lead. We observed that directly training an aggregator yields limited gains, particularly when facing inconsistent candidate answers, i.e., a challenge that directly motivated our proposed curriculum learning-guided RL strategy. This confirms that our method’s effectiveness stems from the process-level collaboration of specialized agents, which captures domain-specific nuances that cannot be replicated by simply aggregating generalist outputs. We have included these additional experiments in Appendix G.12.
> > >
> > > **Table R1**:  Performance comparison with GPT-4o+an aggregator (based on Qwen2.5-VL 7B) baselines.
> > >
> > > | Model | VQA-RAD | SLAKE | PathVQA | OmniMedVQA | MMMU-Med | Overall |
> > > | :--- | :---: | :---: | :---: | :---: | :---: | :---: |
> > > | Qwen2.5-VL-7B (Base) | 61.8 | 64.7 | 60.5 | 60.8 | 56.6 | 60.9 |
> > > | GPT-4o | 61.0 | 75.5 | 69.4 | 68.5 | 69.7 | 68.8 |
> > > | GPT-4o+Qwen2.5-VL-7B (SFT Fine-tuned) | 65.8 | 69.4 | 62.9 | 60.3 | 65.3 | 64.7 |
> > > | GPT-4o+Qwen2.5-VL-7B (GRPO Fine-tuned) | 67.3 | 70.8 | 63.8 | 61.8 | 64.6 | 65.7 |
> > > | MMedAgent-RL | **71.5** | **76.2** | **72.3** | **73.3** | **71.9** | **73.0** |

---

> > > > ### Comment · Reviewer_NzMH · 2025-11-26
> > > >
> > > > I appreciate the author's response; my doubts have been largely resolved.
> > > >
> > > > However, I found that many potentially highly relevant works have not been discussed or mentioned, including, but certainly not limited to, the following papers:
> > > >
> > > > 1. Tang, X., Shao, D., Sohn, J., Chen, J., Zhang, J., Xiang, J., ... & Gerstein, M. (2025). Medagentsbench: Benchmarking thinking models and agent frameworks for complex medical reasoning. arXiv preprint arXiv:2503.07459.
> > > > 2. Fan, Z., Wei, L., Tang, J., Chen, W., Siyuan, W., Wei, Z., & Huang, F. (2025, January). Ai hospital: Benchmarking large language models in a multi-agent medical interaction simulator. In Proceedings of the 31st International Conference on Computational Linguistics (pp. 10183-10213).
> > > > 3. Feng, Y., Wang, J., Zhou, L., Lei, Z., & Li, Y. (2025). Doctoragent-rl: A multi-agent collaborative reinforcement learning system for multi-turn clinical dialogue. arXiv preprint arXiv:2505.19630.
> > > > 4. Schmidgall, S., Ziaei, R., Harris, C., Reis, E., Jopling, J., & Moor, M. (2024). AgentClinic: a multimodal agent benchmark to evaluate AI in simulated clinical environments. arXiv preprint arXiv:2405.07960.
> > > > 5. Zhang, X., Wang, Y., Feng, Z., Chen, R., Zhou, Z., Zhang, Y., ... & Liu, Z. (2025). Med-U1: Incentivizing Unified Medical Reasoning in LLMs via Large-scale Reinforcement Learning. arXiv preprint arXiv:2506.12307.
> > > >
> > > > In particular, I would like the authors to further discuss and clarify the relationship between [3-5] and this paper, and whether they can be compared.
> > > >
> > > > Nevertheless, as I said, I have raised my score to 6.

---

> > > > > ### Author Response · Authors · 2025-11-26
> > > > >
> > > > > Thank you for your thoughtful review. We are glad that our responses could address your concerns, and it is our honor to receive your positive feedback. We have incorporated all suggested references [1-5] into the revised Related Work.

---

> ### Author Response · Authors · 2025-11-21
> **Response to Reviewer NzMH (3/3)**
>
> >**Q5**: Can you provide results progressively adding components (base → +triage → +multi-expert → +curriculum RL) to quantify each contribution?
>
> **A5**: We apologize for the omission of this ablation study.
> To quantify the contribution of each component, we have conducted a progressive evaluation as shown in **Table R4**. Regarding the order of ablation, we formulated the progression as Base → Multi-expert → Triage. From an architectural perspective, the Triage module depends on the existence of a candidate pool of experts to perform routing. Thus, we first introduce the Specialists (Multi-expert) to build the capability pool, and subsequently add the Triage module to manage and utilize these experts efficiently.
> Starting with the Qwen2.5-VL-7B baseline, we observed the following trends:
>
> + Specialists: Integrating domain-specific experts (with a base model as the router) yields further gains, surpassing the single model with voting.
>
> + Triage: Introducing the learned Triage module significantly improves the effective utilization of specialists.
>
> + Curriculum RL: Finally, applying our Curriculum RL strategy provides the most substantial performance leap, demonstrating that optimizing the collaboration between the triage and specialist agents is critical for complex medical reasoning.
>
> We have added this breakdown to the revised manuscript in Appendix G.7 to clearly validate the necessity of each module.
>
> **Table R4**: Performance progressively adding components.
> | Method | VQA-RAD | SLAKE | PathVQA | OmniMedVQA | MMMU-Med |
> | :--- | :---: | :---: | :---: | :---: | :---: |
> | Qwen2.5-VL-7B (Base) | 61.8 | 64.7 | 60.5 | 60.8 | 56.6 |
> | + Specialists (Base Model) | 64.5 | 66.9 | 63.2 | 63.4 | 60.7 |
> | + Triage | 65.7 | 68.4 | 64.4 | 64.8 | 62.6 |
> | + Curriculum RL | **71.5** | **76.2** | **72.3** | **73.3** | **71.9** |

---

### Author Response · Authors · 2025-11-21
**Summary of Paper Revision**

We sincerely appreciate all reviewers for their insightful and constructive feedback. According to these comments, we have improved the paper (new pdf uploaded) and highlighted the main changes with blue text. Below, we summarize all changes:

- We have added comprehensive ablation studies using open-source models as specialists in Appendix G.2. (Reviewers NzMH, noLG, and A9FJ)
- We have included comparisons with test-time scaling techniques in Appendix G.4. (Reviewer NzMH)
- We have conducted a targeted evaluation on hard samples where all specialists fail to quantify the model's correction capability in Appendix G.5. (Reviewer NzMH)
- We have added a progressive ablation study to quantify the contribution of each component in Appendix G.7. (Reviewer NzMH)
- We have added a comparison experiment on GP Update Strategies (independently and simultaneously) in Appendix G.10. (Reviewer dwrJ)
- We have included a human expert evaluation to benchmark diagnostic accuracy and reasoning quality in Appendix G.9. (Reviewer dwrJ)
- We have added comparisons of different Triage Agent optimization strategies in Appendix G.6 and G.8. (Reviewers noLG and A9FJ)
- We have demonstrated the framework's transferability by applying it to the InternVL2.5 architecture in Appendix G.11. (Reviewer dwrJ)
- We have clarified the definitions of SFT baselines and the experimental setup for decision difficulty in Sections 5.1, 5.3, and Appendix G.1. (Reviewers noLG and A9FJ)

---

### Author Response · Authors · 2025-11-30
**Summary of Rebuttal Updates and Reviewer Consensus (Paper ID: 7891)**

Dear Program Chairs, Senior Area Chairs, Area Chairs, and Reviewers:

We sincerely thank you for your time and dedication throughout the review process. We are particularly grateful for the constructive feedback from all reviewers (`NzMH, A9FJ, dwrJ, noLG`), which has significantly helped us strengthen our manuscript.

We understand that the recent OpenReview system rollback may have affected the visibility of the latest discussions and score updates. To assist in your final assessment, we would like to provide a factual summary of the consensus reached and the specific resolutions provided during the rebuttal period.

During the discussion phase, we engaged in active dialogue with the reviewers, resulting in the following developments:

* **Reviewer NzMH**: Explicitly acknowledged our new experiments and stated on `Nov 26`: *"I will raise my score to 6."*
* **Reviewer A9FJ**:  Actively engaged with our response, subsequently confirming that the concerns were resolved and retaining their positive ratings on `Nov 26`.
* **Reviewer noLG and dwrJ**:  We provided a comprehensive response to the concerns. Although we have not yet received a final reply, we believe our revisions effectively address the questions raised.

To provide a clear overview for your decision-making, we summarize the key technical improvements and clarifications made in response to the reviewers:

1. Addressed Concerns on Baselines & Necessity (**Reviewer NzMH, Score 4 → 6**)
* We added comparisons with test-time scaling strategies (Majority Voting, Self-Consistency) and the trained aggregator [Appendix G.4 and G.12].
* We evaluated performance on hard samples where all specialists fail [Appendix G.5].
* We demonstrated that our framework remains effective using open-source specialists [Appendix G.2].

****
2. Addressed Concerns on Methodology & Implementation (**Reviewer A9FJ, Score 6 → 6**)
* We clarified the SFT baselines and quantitatively demonstrated that our GRPO-based approach significantly outperforms standard SFT [Appendix G.1].
* We confirmed that our difficulty grading is strictly used for curriculum training and post-hoc analysis, ensuring no data leakage during inference [Appendix F.4].
****
3. Addressed Concerns on Clinical Alignment & Validity (**Reviewer dwrJ, Score 8**)
* We aligned our workflow with established medical AI paradigms and conducted a human expert evaluation [Appendix G.9].
* We validated our framework on a different backbone (InternVL2.5), showing consistent gains [Appendix G.11].
****
4. Addressed Concerns on Visualization & Technical Details (**Reviewer noLG, Score 4**)
* We have revised *Figure 1* to explicitly include the General Practitioner and multi-agent module, resolving the ambiguity regarding the practical workflow.
* We provided new quantitative evidence [Appendix G.8] showing that GRPO-optimized triage yields superior generalization compared to Base and SFT models.
* We elaborated on the dynamic entropy regularization mechanism used to balance exploration and exploitation in Section 3.3.
****
We hope this summary assists you in navigating the discussion history, especially given the system interruptions. We remain fully committed to incorporating all suggestions into the final version of the paper.

Thank you again for your time and consideration.

Best regards,

Authors of Paper 7891

---

### Meta-Review · Area_Chair_g5GA · 2026-01-07

**Summary:**

Most concerns are on the reliance on close-source models (*e.g.,* the o3 family) and missed analysis and issues with clarity. The authors responded with additional experiments with open-source models as well as additional analysis/revisions.

**Reviewer Concerns:**

`NzMH`:
Generally addressed: Missing baselines with simpler baselines, no evidence on performance gains on key components, privacy concerns over proprietary models.


`noLG`:
Generally addressed: Clarity in illustrations; missing discussions on the triage component’s influence; choice of o3 for specialities; clarity in experiments



`dwrJ`:
Generally addressed: Lack of human experts involvement; some details missing
Not fully addressed: Alignment with the actual clinical flow


`A9FJ`:
Reviewer responded before Nov. 27 confirming most questions addressed.

**Reviewer Scores:**

Most of `noLG`’s concerns are regarding clarity / missed analysis which seem to be reasonably addressed over the rebuttal. Other reviewers were positive before Nov 27.

---

### Decision · Program_Chairs · 2026-01-26

Accept (Poster)